# Deep Neural Network Regression with Functional Covariates

**Hang Zhou** [1 2]  **Ju-Sheng Hong** [3]  **Xiucai Ding** [3]  **Jane-Ling Wang** [3]

## Abstract

Regression with functional covariates poses fundamental challenges due to the infinite-dimensional nature of functional data, and its theoretical properties have been studied under specialized frameworks in classical nonparametric statistics. While deep neural networks (DNNs) have demonstrated remarkable empirical success in high-dimensional regression, their theoretical behavior in settings involving infinite-dimensional covariates remains largely unexplored. In this work, we study the theoretical performance of DNN-based estimators for regression problems with functional covariates. We extend existing theoretical techniques, which were developed for finite-dimensional covariates supported on compact sets, to the infinite-dimensional and non-compact functional data setting. Under mild conditions, we show that DNN estimators attain minimax-optimal polynomial rates of convergence for both functional linear models and functional generalized linear models. For fully nonparametric regression with functional covariates, we establish a lower bound on the prediction error and further discuss the fundamental obstacles inherent to this problem and their connections to existing state-of-the-art methods in the literature.

## 1. Introduction

Many datasets arising from scientific studies and daily life are naturally viewed as realizations of smooth curves, such as daily temperature records, growth trajectories, or longitudinal biomedical signals. Functional data analysis (FDA) provides a framework for modeling and analyzing such data by treating the objects of interest as random functions or curves. Over the past several decades, FDA has developed into a mature field, with a rich body of methods successfully applied across diverse domains, including medicine, biology, brain imaging, and economics. For a comprehensive overview, we refer to the review paper by Wang et al. (2016) and monographs by Ramsay & Silverman (2006); Ferraty & Vieu (2006); Horváth & Kokoszka (2012); Hsing & Eubank (2015).

Functional data have two key characteristics: smoothness and infinite dimensionality. In particular, smoothness is characterized by the decay rate of the eigenvalues of the covariance operator. Infinite dimensionality implies that functional data cannot be fully represented by a finite number of eigencomponents. This makes the covariance operator non-invertible, creating significant challenges in regression models involving functional covariates. Although infinite dimensionality might initially appear to be a curse, the inherent smoothness of functional data enables consistent estimation of the regression function using the first $K_n$ principal components, provided $K_n$ increases to infinity as the sample size grows. A detailed discussion of regression models with functional covariates is provided in Section 2.1.

Deep neural networks have recently emerged as a popular research topic due to their strong empirical performance across various applications. A notable feature of DNNs is their capability to learn intrinsic relationships between covariates and responses, effectively mitigating the curse of dimensionality when these relationships exhibit low-dimensional structures. An important question that naturally arises is whether DNNs can provide similar benefits when dealing with infinite-dimensional functional data as covariates. This specific problem has only been explored sparingly, and existing theoretical methods cannot be directly applied due to the inherent infinite dimensionality of functional data. Furthermore, the unbounded nature of functional data presents additional challenges, as most current theoretical results for DNNs assume that covariates are supported on compact domains.

In this paper, we investigate the theoretical performance of deep neural networks in regression problems with functional covariates. We begin with the classical setting of functional

---

[1]Department of Statistics and Operations Research, The University of North Carolina at Chapel Hill, Chapel Hill, NC, USA [2]School of Data Science and Society, The University of North Carolina at Chapel Hill, Chapel Hill, NC, USA [3]Department of Statistics, University of California, Davis, Davis, CA, USA. Correspondence to: Jane-Ling Wang <janelwang@ucdavis.edu>.

*Proceedings of the 43^{rd} International Conference on Machine Learning*, Seoul, South Korea. PMLR 306, 2026. Copyright 2026 by the author(s).

linear regression, using it as a benchmark to assess whether deep neural networks can achieve convergence rates comparable to those established in the existing literature. We show that, under mild regularity conditions, the proposed estimators attain the same optimal polynomial rates of convergence, up to a logarithmic factor. We then extend our analysis to functional generalized linear models with known link functions, also referred to as functional nonlinear regression models, and demonstrate that optimal polynomial convergence rates remain achievable in this broader setting. Finally, we consider the more challenging problem of fully nonparametric regression with functional covariates. In this case, we establish a lower bound on the prediction error, showing that polynomial rates of convergence are generally unattainable without additional structural assumptions. We further discuss the inherent difficulties of this problem and the existing state-of-the-art techniques in the literature.

In the rest of this paper, we provide an introduction and literature review on functional data and deep neural network regression in Section 2. The methodologies and main theoretical results are presented in Section 3. Section 4 presents real-world data applications and summarizes key findings from our simulation studies. Proofs and detailed simulation designs are provided in the Appendix.

## 2. Background and Literature Review

### 2.1. Regression Models with Functional Covariates

In functional data analysis, the variables of interest are typically random functions that satisfy certain smoothness conditions. A common modeling approach is to treat these functions as realizations of random elements taking values in a Hilbert space (Hsing & Eubank, 2015). In this paper, we work within the Hilbert space $\mathcal{L}^2[0,1]$, consisting of square-integrable functions on the interval $[0,1]$. Let $X(t)$ be a random element in $\mathcal{L}^2[0,1]$ satisfying $\mathbb{E}[\{X(t)\}^2] < \infty$ for all $t \in [0,1]$. The mean function $\mu(t) = \mathbb{E}[X(t)]$ and the covariance function $C(s,t) = \mathbb{E}[(X(s)-\mu(s))(X(t)-\mu(t))]$ are then well-defined.

Regression models are fundamental tools for understanding relationships between a response variable and its predictors. In this paper, we focus on the scalar-on-function regression setting, where the response $Y$ is scalar and the predictor $X$ is a functional covariate. Numerous regression approaches have been proposed to investigate the relationship between $Y$ and $X$. A foundational model is the *functional linear regression* model, which is an extension of multivariate linear regression and is defined as:

$$Y = \int_0^1 \{X(t) - \mu(t)\}\beta(t)\,\mathrm{d}t + \varepsilon, \qquad (1)$$

where $\varepsilon$ denotes the regression error, and $\beta(t) \in \mathcal{L}^2[0,1]$

is the regression function, also known as the slope function. The seminal work Hall & Horowitz (2007) shows that the principal component analysis based estimator of the slope function attains a nonparametric convergence rate determined by the smoothness properties of both $X(t)$ and $\beta(t)$, which is optimal in the minimax sense. Additionally, Cai & Hall (2006) shows that the prediction error of the estimator from Hall & Horowitz (2007) converges faster than its estimation error, yet remains slower than the parametric $\sqrt{n}$-rate. In a parallel line of research, Yuan & Cai (2010) and Cai & Yuan (2012) derive analogous results within the reproducing kernel Hilbert space framework.

Beyond linear models, various nonlinear regression approaches have been developed to study relationships between scalar responses and functional covariates. For example, in functional generalized linear models, Müller & Stadtmüller (2005) established the asymptotic normality of slope estimators, while Dou et al. (2012) introduced a change-of-measure technique and derived the minimax optimal rate for estimation errors. Additionally, Qu et al. (2016) studied estimation errors in functional Cox models with censored data. In these works, although the relationships between $X$ and $Y$ are nonlinear, certain nonlinear components, such as the link function, are typically assumed to be known. A more general framework, in which the nonlinear link function is unknown, leads to the challenging scenario of functional nonparametric regression. Ferraty & Vieu (2006) showed that, in this setting, polynomial convergence rates cannot be achieved in general, and only logarithmic rates are attainable under strong conditions. This difficulty arises from the inherently infinite-dimensional nature of functional data and the consequences of low small-ball probabilities (Li & Linde, 1999; Dabo-Niang, 2002). Given these challenges, researchers have focused on incorporating additional structural assumptions into functional regression models. One successful approach is the functional single-index model studied by Jiang & Wang (2011). In this setting, Chen et al. (2011) demonstrated that polynomial convergence rates for estimation errors are achievable.

### 2.2. Deep Neural Network Regression

Deep neural networks have been successfully demonstrated to yield impressive results in solving complex regression and classification problems, especially when the covariates are high-dimensional. Suppose we observe i.i.d. random variables $(\boldsymbol{X}_i, Y_i)$ generated by

$$Y_i = f(\boldsymbol{X}_i) + \varepsilon_i, \quad i = 1, \ldots, n, \qquad (2)$$

where $\boldsymbol{X}_i \in \mathbb{R}^d$ is the $d$-dimensional covariate and $f$ is an unknown regression function. We assume that the regression errors $\varepsilon_i$ have mean zero and a finite variance $\sigma^2$. The regression function is typically estimated by minimizing the following empirical loss function over a certain function

class $\mathcal{F}$:

$$\hat{f} = \arg\min_{\tilde{f} \in \mathcal{F}} \frac{1}{n} \sum_{i=1}^{n} \{Y_i - \tilde{f}(\boldsymbol{X}_i)\}^2. \qquad (3)$$

When the class $\mathcal{F}$ is chosen as the set of neural networks, defined as

$$\mathcal{F} = \{f(\mathbf{x}) = L_D \circ \sigma \circ L_{D-1} \circ \sigma \circ \cdots \circ \sigma \circ L_1(\mathbf{x})\} \quad (4)$$

with $L_i(\mathbf{x}) = W_i\mathbf{x} + \mathbf{b}_i$, weight matrices $W_i$, bias vectors $\mathbf{b}_i$, a nonlinear activation function $\sigma$, and depth $D$, the resulting estimator from (3) is referred to as the DNN estimator. Recently, several studies have investigated the convergence rate of the prediction error for the deep neural network regression estimator under various choices of the network class $\mathcal{F}$ (Schmidt-Hieber, 2020; Kohler & Langer, 2021).

If the regression function $f(\cdot)$ in (2) is Hölder smooth with smoothness parameter $\alpha$, the classical minimax nonparametric convergence rate is known to be $n^{-2\alpha/(2\alpha+d)}$ (Stone, 1982). This rate is well known to suffer from the curse of dimensionality, deteriorating rapidly as the dimension $d$ increases. Deep neural networks have demonstrated their practical effectiveness in dimension reduction and in capturing intrinsic relationships between the covariate $\boldsymbol{X}$ and the response $Y$. Numerous studies have investigated the theoretical foundations underpinning these capabilities, primarily through two main lines of research. The first line involves incorporating additional structural assumptions on the regression function $f(\cdot)$. Examples include considering compositional function classes of smooth functions (Bauer & Kohler, 2019; Schmidt-Hieber, 2020), anisotropic function spaces (Suzuki & Nitanda, 2021; Okumoto & Suzuki, 2022), and function spaces with mixed smoothness (Suzuki, 2019). The second line of research focuses on making specific distributional assumptions about the covariates $\boldsymbol{X}$. Examples in this category include assuming that the covariates reside on manifolds (Schmidt-Hieber, 2019; Chen et al., 2019; Jiao et al., 2023), or that the support of $\boldsymbol{X}$ has low Minkowski dimension (Nakada & Imaizumi, 2020; Zhang et al., 2023). Both approaches lead to improved convergence rates of the form $n^{-2\alpha/(2\alpha+d^*)}$, where $d^* \ll d$ represents either the intrinsic dimension of the regression function or the intrinsic dimension of the covariate $\boldsymbol{X}$.

## 3. Deep Regression with Functional Covariates

In this section, we consider the regression model (2) with a functional covariate and investigate the convergence rate of the prediction error for deep neural network estimators in different regression settings. Specifically, we assume that the observed data $\{(X_i, Y_i)\}_{i=1}^{n}$ are independent and identically distributed samples from the model $Y_i = f(X_i) + \varepsilon_i$, where $X_i \in \mathcal{L}^2[0,1]$ and the errors $\varepsilon_i$ are zero-mean random variables with finite variance. To facilitate a clearer understanding of the problem and its associated challenges, we first examine the simplest scenario, functional linear regression, to assess whether deep neural networks can achieve the classical nonparametric convergence rate. Next, we extend our discussion to nonlinear regression models with a known link function, where the objective is to estimate the regression slope function. Finally, we explore the most challenging scenario, functional nonparametric regression, where the nonlinear relationship between the response and the functional covariate is entirely unknown.

Since neural networks take vector inputs rather than functions, we first discretize the functional covariates $X_i$ into finite-dimensional vectors to construct the neural network regression estimator. Because $X_i$ is a second-order stochastic process, Mercer's theorem (Indritz, 1963) ensures that the covariance function $C(s,t)$ admits a spectral decomposition of the form $C(s,t) = \sum_{k=1}^{\infty} \lambda_k \phi_k(s) \phi_k(t)$, where $\lambda_k$ are eigenvalues and $\phi_k(\cdot)$ are the associated eigenfunctions. Thus, the Karhunen–Loève expansion allows us to represent each functional covariate as

$$X_i(t) = \mu(t) + \sum_{k=1}^{\infty} \xi_{ik} \phi_k(t), \qquad (5)$$

where the principal component scores $\{\xi_{ik}\}_{k=1}^{\infty}$ are uncorrelated random variables with zero mean and variances $\mathbb{E}[\xi_{ik}^2] = \lambda_k$. Consequently, the process $X_i(t)$ is equivalent to an infinite-dimensional vector representation $(\xi_{i1}, \xi_{i2}, \ldots, \xi_{ik}, \ldots)$. The principal component decomposition captures two critical properties of functional data: infinite dimensionality and inherent smoothness. On one hand, each $X_i(t)$ resides in an infinite-dimensional Hilbert space; thus it cannot be exactly represented by any finite-dimensional vector. On the other hand, the covariance function $C(s,t)$ satisfies certain smoothness conditions, typically implying polynomial decay rates of the eigenvalues, such as $\lambda_k \asymp k^{-a}$ for some constant $a > 1$. To approximate this infinite-dimensional representation, we truncate the principal component series and use the first $K_n$ scores, $\boldsymbol{\xi}_i = (\xi_{i1}, \xi_{i2}, \ldots, \xi_{iK_n})^{\top}$, as input vectors for the neural network. The regression estimator is then obtained by solving

$$\hat{f} = \arg\min_{\tilde{f} \in \mathcal{F}} \frac{1}{n} \sum_{i=1}^{n} \{Y_i - \tilde{f}(\boldsymbol{\xi}_i)\}^2. \qquad (6)$$

We evaluate the accuracy of the resulting estimator $\hat{f}$ through the prediction error defined as

$$\mathfrak{R}(\hat{f}, f) = \mathbb{E}_* \left[ \{f(X^*) - \hat{f}(\boldsymbol{\xi}^*)\}^2 \right], \qquad (7)$$

where $X^*$ is an independent copy of $X_i$, $\boldsymbol{\xi}^* = (\xi_1^*, \xi_2^*, \ldots, \xi_{K_n}^*)^{\top}$ with $\xi_j^* = \langle X^* - \mu, \phi_j \rangle$, and $\mathbb{E}_*$ denotes the expectation taken over $X^*$ (and $\boldsymbol{\xi}^*$) only. We shall make the following assumptions on the functional covariate.

**Assumption 3.1.** The eigenvalues $\{\lambda_k\}_{k=1}^{\infty}$ satisfy $\lambda_k \asymp k^{-a}$ for some constant $a > 1$.

**Assumption 3.2.** The principal component scores $\xi_k$ are sub-exponential uniformly in $k = 1, \ldots, K_n$.

Assumption 3.1 is a standard polynomial eigenvalue decay condition commonly used in functional principal component analysis (Cai & Hall, 2006; Hall et al., 2006; Dou et al., 2012; Zhou et al., 2025). Exponentially decaying eigenvalues are of less interest in practice, since the number of eigenfunctions that can be accurately estimated is only of order $\log n$, which is rather restrictive. The assumption of sub-exponential tails for the principal component scores is a technical condition used to manage the inherent unboundedness of functional data. Assumption 3.2 can be replaced by alternative tail conditions on the principal component scores without affecting the main results.

### 3.1. Functional Linear Regression

For the classical multivariate linear regression model $Y = \boldsymbol{X}^{\top}\beta + \varepsilon$, the normal equation yields the solution $\beta = (\mathbb{E}[\boldsymbol{X}\boldsymbol{X}^{\top}])^{-1}\mathbb{E}[\boldsymbol{X}Y]$, where $\mathbb{E}[\boldsymbol{X}\boldsymbol{X}^{\top}]$ is the population covariance matrix. However, in the functional linear regression (1), the covariate is an infinite-dimensional stochastic process; thus its covariance operator is non-invertible. To address this issue, regularization is needed, and functional principal component analysis (FPCA) is a fundamental tool (Cai & Hall, 2006; Hall & Horowitz, 2007; Dou et al., 2012). Specifically, FPCA regularization involves truncating the infinite-dimensional representation to the first $K_n$ principal components to construct a feasible regression estimator. Due to the infinite-dimensional nature of functional data, achieving consistency requires that $K_n$ grow with the sample size $n$. In this scenario, the estimation variance increases as $K_n$ grows, while the truncation bias decreases. Thus, an optimal choice of $K_n$ that balances these two aspects leads to the optimal convergence rate. A similar consideration applies to our deep neural network estimator in (6). Here, the number of principal components $K_n$ must also increase with the sample size to guarantee consistency. However, most existing theoretical results for neural network regression focus on fixed-dimensional input spaces and thus cannot be directly applied to the scenario considered here, where the input dimension grows with the sample size.

In functional linear regression, analyzing the convergence rate typically requires two key assumptions, as it depends on the smoothness properties of both the covariate process $X$ and the regression function $\beta$. In addition to the common assumption on the eigenvalues, a smoothness condition on the regression function $\beta$ is also needed. A widely adopted smoothness assumption is that the Fourier coefficients of $\beta$ with respect to the eigenfunctions $\{\phi_k\}_{k=1}^{\infty}$ of the covariance operator also follow a polynomial decay rate. That is, if $\beta$ admits the expansion $\beta(t) = \sum_{k=1}^{\infty} b_k \phi_k(t)$, we assume $b_k \asymp k^{-b}$ for some constant $b > a/2 + 1$ (Cai & Hall, 2006; Hall & Horowitz, 2007). Under these assumptions, the $\mathcal{L}^2$-estimation error for the FPCA-based estimator achieves the convergence rate $n^{-(2b-1)/(a+2b)}$ (Hall & Horowitz, 2007), while the corresponding prediction error achieves the faster rate $n^{-(a+2b-1)/(a+2b)}$ (Cai & Hall, 2006). Both rates are optimal in the minimax sense. Since deep neural networks are primarily designed for prediction tasks, our analysis focuses specifically on the prediction error. Evaluating whether deep neural network regression estimators can attain this optimal prediction error rate in functional linear regression is crucial for understanding the role of deep neural networks in nonparametric regression, which is an area that remains unexplored. The following assumption sets up a function class for $\beta$ with certain smoothness characterized by the decay rates of the Fourier coefficients of $\beta$ projected onto the eigenfunction $\phi_k$. Assumptions 3.1 and 3.3 provide a natural theoretical framework for analyzing convergence rates with functional covariates and are widely adopted in the literature (Cai & Hall, 2006; Hall & Horowitz, 2007; Dou et al., 2012; Zhou et al., 2023).

**Assumption 3.3.** Assume that $\{(X_i, Y_i)\}_{i=1}^{n}$ are i.i.d. samples from the functional linear model (1). The slope function $\beta$ admits the expansion $\beta(t) = \sum_{k=1}^{\infty} b_k \phi_k(t)$, where $\{\phi_k\}_{k=1}^{\infty}$ are the corresponding eigenfunctions and the coefficients satisfy $b_k \asymp k^{-b}$ for some $b > 1/2$.

**Theorem 3.4.** *Under Assumptions 3.1 to 3.3, let $\hat{f}$ be the estimator in (6) where $\mathcal{F}$ is taken to be a fully connected ReLU neural network,*

$$\mathfrak{R}(\hat{f}, f) = \widetilde{O}_P\left(n^{\frac{1-a-2b}{a+2b}}\right),$$

*where $\widetilde{O}_P(\cdot)$ hides logarithmic factors.*

The proof follows the strategy employed in Lemma 4 of Schmidt-Hieber (2020), but significant modifications are necessary because Lemma 4 in Schmidt-Hieber (2020) cannot be directly applied in our context. Specifically, Lemma 4 assumes, first, that the true regression function is bounded and, second, that the domain of the predictors is compact. However, due to the inherently unbounded nature of functional data, the principal component scores $\xi_k^*$ generally have non-compact domains. Consequently, the function $f_0(\boldsymbol{\xi}^*) = \sum_{k=1}^{K_n} \xi_k^* b_k$, which also minimizes $\mathbb{E}[\{f(\boldsymbol{\xi}^*) - \int X^* \beta\}]^2$ over $\mathcal{F}$, does not fit directly within the assumptions of Lemma 4. To address this issue, we extend the proof framework of Lemma 4 by introducing a truncation step for the principal component scores $\xi_k^*$, which is a standard technique in functional data analysis (Zhang & Wang, 2016; Zhou et al., 2025). Another challenge arises from the truncation itself, namely the additional bias introduced by truncation, given by $\sum_{k=K_n+1}^{\infty} \xi_k^* b_k$, which requires further analysis. Finally, the convergence rate is

derived by selecting an optimal truncation parameter $K_n$ that balances the trade-off between estimation variance and truncation bias.

It is instructive to compare the convergence rate established in Theorem 3.4 with the classical results in Cai & Hall (2006). Both rates exhibit polynomial decay with respect to the sample size $n$ and, importantly, share the same polynomial exponent. However, the convergence rate in Theorem 3.4 contains an additional $\log n$ factor, making it slightly slower than the rate obtained in Cai & Hall (2006). Despite this, our result is derived under substantially weaker assumptions in two important respects. First, we do not impose the condition $\mathbb{E}(\xi_{j_1}\xi_{j_2}\xi_{j_3}\xi_{j_4}) = 0$ unless indices are repeated, a technical assumption commonly adopted in functional principal component analysis. Second, we relax the restrictive smoothness requirement $b \geq a + 2$ on the regression function $\beta$, which is typically needed to control perturbation series expansions of eigenfunctions. In summary, by leveraging the flexibility and expressive power of deep neural networks, we obtain convergence rates that are nearly identical to those of classical methods, differing only by a logarithmic factor, while significantly relaxing their restrictive technical assumptions.

### 3.2. Functional Nonlinear Regression

In this subsection, we extend the functional linear regression (1) to generalized functional models with known link functions. The relevant literature dates back to Müller & Stadtmüller (2005), which first investigated convergence rates for functional generalized linear models, assuming that the bias due to approximating the infinite-dimensional model by finite-dimensional sequences is negligible. The minimax optimality for functional generalized linear models was later studied by Dou et al. (2012), who introduced a change-of-measure technique to rigorously address model approximation issues and establish minimax optimal results. We assume that the mean function $\mu = 0$ and

$$Y_i \mid X_i \sim Q_{\lambda_i}, \quad \text{with } \lambda_i = f(X_i) = \int X_i\beta, \quad (8)$$

where $Q_\lambda$ belongs to a one-parameter exponential family with density function

$$Q_\lambda = h(y)\exp\{\lambda y - \psi(\lambda)\},$$

where $h(y)$ is the base measure, $\psi(0) = 0$, and the distribution $Q_\lambda$ has mean $\psi'(\lambda)$ and variance $\psi''(\lambda)$. We make the following assumptions to facilitate bounding the prediction error, which are the same as those used in Dou et al. (2012).

**Assumption 3.5.** Assume that $\{(X_i, Y_i)\}_{i=1}^n$ are i.i.d. samples from the functional generalized linear model (8). The slope function $\beta$ admits the expansion $\beta(t) = \sum_{k=1}^\infty b_k\phi_k(t)$, where $\{\phi_k\}_{k=1}^\infty$ are the corresponding eigenfunctions and the coefficients satisfy $b_k \asymp k^{-b}$ for some $b > 1/2$.

**Assumption 3.6.** For each $\epsilon > 0$, there exists a finite constant $C_\epsilon$ such that $\psi''(\lambda) \leq C_\epsilon \exp(\epsilon\lambda^2)$.

To accommodate a wide range of response distributions, generalized linear models estimate parameters by maximizing the log-likelihood rather than minimizing the least squares criterion. In functional generalized linear models, Dou et al. (2012) proposed truncating the functional predictor to the first $K_n$ principal component scores and then maximizing the corresponding approximated likelihood function. Similar to functional linear models, $K_n$ must grow with the sample size to eliminate truncation bias. Recently, Meng & Li (2026) applied deep neural networks to estimate multivariate generalized linear models by minimizing the negative log-likelihood. We adopt and extend this approach to the scenario of growing dimensionality, which naturally aligns with the functional data setting. Specifically, the estimator of the natural-parameter function is defined by minimizing the negative log-likelihood function:

$$\hat{f} = \arg\min_{f\in\mathcal{F}} \frac{1}{n}\sum_{i=1}^n -y_i f(\boldsymbol{\xi}_i) + \psi(f(\boldsymbol{\xi}_i)). \quad (9)$$

While the $\mathcal{L}^2$ prediction error in (7) is natural for Gaussian linear models, it is generally not the most appropriate loss for generalized linear models. For exponential-family responses with density $h(y)\exp\{\lambda y - \psi(\lambda)\}$, we measure the discrepancy between the predicted natural parameter $\hat{f}(\boldsymbol{\xi}^*)$ and the true natural parameter $f(X^*)$ using the Bregman divergence induced by the cumulant function $\psi$:

$$\mathcal{B}_\psi\left(\hat{f}(\boldsymbol{\xi}^*), f(X^*)\right) := \psi\left(\hat{f}(\boldsymbol{\xi}^*)\right) - \psi\left(f(X^*)\right)$$
$$- \psi'\left(f(X^*)\right)\left\{\hat{f}(\boldsymbol{\xi}^*) - f(X^*)\right\}.$$

This quantity is the conditional excess negative log-likelihood. Moreover, by Taylor's theorem, for some $c^*$ between $\hat{f}(\boldsymbol{\xi}^*)$ and $f(X^*)$,

$$\mathcal{B}_\psi\left(\hat{f}(\boldsymbol{\xi}^*), f(X^*)\right) = \frac{1}{2}\psi''(c^*)\left\{\hat{f}(\boldsymbol{\xi}^*) - f(X^*)\right\}^2.$$

In particular, for the Gaussian linear model $\psi(u) = u^2/2$, $\mathcal{B}_\psi(u, v) = (u - v)^2/2$ and hence $\mathbb{E}_*[\mathcal{B}_\psi(\hat{f}(\boldsymbol{\xi}^*), f(X^*))] = \mathbb{E}_*[\{f(X^*) - \hat{f}(\boldsymbol{\xi}^*)\}^2]/2 = \mathfrak{R}(\hat{f}, f)/2$.

Similar to the functional linear regression case, the following theorem shows that the proposed DNN estimator for the functional generalized linear model achieves the minimax-optimal convergence rate for the prediction error, up to a logarithmic factor in $n$. The proof follows the same general framework as that of Theorem 3.4, with additional technical

arguments required to handle the nonlinear link function in the generalized linear model. To the best of our knowledge, this is the first result establishing the optimal convergence rate for functional generalized linear models using DNN estimators, and it provides concrete insight into how dimensionality reduction of functional covariates influences the learning behavior of deep neural networks.

**Theorem 3.7.** *Under Assumptions 3.1, 3.5 and 3.6, and assume in addition that $\{\xi_k\}_{k \leq K_n}$ are sub-Gaussian uniformly in $k$, let $\hat{f}$ be defined by* (9) *over a ReLU network class $\mathcal{F}$. Then for $K_n \asymp n^{1/(a+2b)}$ (up to $\log$ factors),*

$$\mathbb{E}_* \left[ \mathcal{B}_\psi(\hat{f}(\boldsymbol{\xi}^*), f(X^*)) \right] = \widetilde{O}_P \left( n^{\frac{1-a-2b}{a+2b}} \right),$$

*where $\widetilde{O}_P(\cdot)$ hides logarithmic factors.*

### 3.3. Functional Nonparametric Regression

In Sections 3.1 and 3.2, we studied the convergence rates for the prediction error of the proposed method under both the functional linear model and the functional generalized linear model, obtaining optimal polynomial rates consistent with those from classical methods. In these models, the relationships between responses and functional covariates are either linear or nonlinear with known link functions. In this subsection, we consider functional nonparametric regression, where the nonlinear relationship between the response and functional covariates is unknown. This is a particularly challenging problem, and the following theorem provides a lower bound for Gaussian processes.

**Theorem 3.8.** *Consider the functional nonparametric regression model $Y_i = f(X_i) + \varepsilon_i$, $i = 1, \ldots, n$, where $\varepsilon_i \overset{\text{i.i.d.}}{\sim} N(0, \sigma^2)$ are independent of $\{X_i\}_{i=1}^n$. Fix $p \in (0, 1]$ and $L > 0$, and define the Hölder class*

$$\mathfrak{H}(L, p) := \left\{ f : \mathcal{L}^2[0,1] \to \mathbb{R} : \sup_{x \neq y} \frac{|f(x) - f(y)|}{\|x - y\|^p} \leq L \right\}.$$

*Let $\hat{f}_n$ be any estimator based on $\{(X_i, Y_i)\}_{i=1}^n$. Assume $X$ is a centered Gaussian random element in $\mathcal{L}^2[0,1]$ whose covariance eigenvalues satisfy Assumption 3.1 with decay parameter $a > 1$. Then there exists a constant $c > 0$ such that for all large $n$,*

$$\sup_{f \in \mathfrak{H}(L,p)} \mathbb{E}_f \left[ \{\hat{f}_n(\mathfrak{o}) - f(\mathfrak{o})\}^2 \right] \geq c \, (\log n)^{-p(a-1)},$$

*where $\mathfrak{o} \in \mathcal{L}^2[0,1]$ denotes the zero function.*

Theorem 3.8 implies $n^\gamma \sup_{f \in \mathfrak{H}(L,p)} \mathbb{E}[(\hat{f}_n(\mathfrak{o}) - f(\mathfrak{o}))^2] \to \infty$ as $n \to \infty$ for every $\gamma > 0$, which is consistent with the general minimax lower bounds for functional nonparametric regression built via small-ball probabilities (Mas, 2012, Theorem 3). This means that, without additional structural

assumptions, polynomial convergence rates generally cannot be achieved under standard conditions (Ferraty & Vieu, 2006; Ferraty et al., 2007). More specialized models, such as functional single-index models, assume that the nonlinear relationship between the response and functional predictor can be expressed through a single index (Chen et al., 2011; Jiang & Wang, 2011). For instance, Chen et al. (2011) derived polynomial convergence rates for the estimation error in functional single-index models by assuming polynomial decay of the tail sums of the Fourier coefficients of $\beta$.

Deep neural networks are known for their ability to perform dimension reduction and uncover intrinsic relationships between responses and predictors. However, extending existing theoretical results to cases with growing dimensionality to achieve polynomial convergence rates is challenging. In most existing DNN literature, dimension reduction relies on imposing smoothness assumptions on the regression function. For example, Schmidt-Hieber (2019) considers function classes composed of Hölder continuous functions, achieving dimension reduction by assuming a low intrinsic dimension of the regression function. However, this approach does not translate directly to functional data contexts. Specifically, consider the functional single-index model given by $Y_i = f(\int X_i \beta) + \varepsilon_i$, which can be rewritten as $Y_i = f(\sum_{k=1}^\infty \xi_{ik} b_k) + \varepsilon_i$. After truncating the principal components, $f(\sum_{k=1}^\infty \xi_{ik} b_k)$ can be approximated by $\tilde{f}(\sum_{k=1}^{K_n} \xi_{ik} b_k)$, represented as a composition function $h(\boldsymbol{\xi}_i) = \tilde{f} \circ \tilde{h}(\boldsymbol{\xi}_i)$, where $\tilde{f} : \mathbb{R} \mapsto \mathbb{R}$ and $\tilde{h} : \mathbb{R}^{K_n} \mapsto \mathbb{R}$, with $\tilde{h}(\boldsymbol{\xi}_i) = \sum_{k=1}^{K_n} \xi_{ik} b_k$ being linear. Since the intrinsic dimensions of $\tilde{f}$ and $\tilde{h}$ are 1 and $K_n$ respectively, the convergence rate of using a DNN to estimate $h(\cdot)$ is $n^{-2\tau/(2\tau+K_n)}$, where $\tau$ denotes the smoothness parameter. This is a typical $K_n$-dimensional nonparametric convergence rate. Because $K_n$ must grow with the sample size to reduce the approximation error between $f(\sum_{k=1}^{K_n} \xi_{ik} b_k)$ and $f(\sum_{k=1}^\infty \xi_{ik} b_k)$, achieving polynomial convergence becomes problematic. Even assuming a polynomial approximation bias for $\mathbb{E}|f(\sum_{k=1}^{K_n} \xi_{ik} b_k) - f(\sum_{k=1}^\infty \xi_{ik} b_k)|^2$ in terms of $K_n$, one can only guarantee logarithmic convergence, because the generalization error rate $n^{-2\tau/(2\tau+K_n)}$ implies that $K_n$ cannot increase too fast.

Besides imposing smoothness or structural assumptions on the regression function $f$, another line of research in the theoretical analysis of DNNs focuses on the geometric properties of the covariate space. In Jiao et al. (2023), the authors studied the regression model $Y = f_0(X) + \varepsilon$, where the covariate $X \in \mathbb{R}^d$ is supported around low-dimensional subsets of $\mathbb{R}^d$. For instance, if $X$ is supported on a compact $d_\mathcal{M}$-dimensional Riemannian manifold embedded in $\mathbb{R}^d$, the convergence rate of the prediction error improves to $n^{-2\tau/(2\tau+d_\mathcal{M})}$ instead of the standard $n^{-2\tau/(2\tau+d)}$, where $\tau$ denotes the smoothness parameter. Another approach

is to characterize the complexity of the support of $X$ using its Minkowski dimension (Nakada & Imaizumi, 2020; Jiao et al., 2023; Zhang et al., 2023). Specifically, Nakada & Imaizumi (2020) obtained convergence rates matching the nonparametric rate of dimension $d^{\#}$, where $d^{\#}$ can be chosen arbitrarily close to the Minkowski dimension of the support of $X$. Moreover, Zhang et al. (2023) extended the notion of Minkowski dimension and analyzed scenarios where $X \sim N(0, \Sigma)$, with $\Sigma = \text{diag}(\gamma_1, \gamma_2, \ldots, \gamma_d)$ and polynomially decaying eigenvalues $\gamma_k$ with decay rate $\omega$. They showed that the effective Minkowski dimension of $X$ is $n^{(1+1/\omega)/\omega}$, leading to faster convergence rates when $n^{(1+1/\omega)/\omega}$ is smaller than the original dimension $d$. Although this setting is similar to regression settings with functional covariates, where $X$ is represented by the first $K_n$ principal component scores, even employing the effective Minkowski dimension does not yield polynomial convergence rates with current analytical techniques. Whether deep neural networks can achieve polynomial convergence rates for functional nonparametric regression remains an open problem. We hope that the methodologies and discussions developed in this work will provide useful guidance for addressing this challenge in future research.

# 4. Numerical Studies

The numerical studies in this paper are designed to illustrate and help readers better understand the theoretical results. Section 3 studies DNN-based regression with functional covariates through a finite-dimensional approximation of the infinite-dimensional functional input, where the number of retained components $K_n$ increases with the sample size $n$. This leads to the key bias–variance trade-off underlying the convergence rates: using too few components produces a large approximation bias, while using too many components increases estimation error. Our simulation studies are therefore constructed to mirror this theoretical setting. In particular, we generate functional covariates from models satisfying the eigenvalue-decay and coefficient-decay assumptions used in Section 3, and we examine how the prediction error changes with both the sample size and the effective dimension of the observed functional input. For synthetic data, where the true regression function is known, we plot $\log(\text{MSE})$ against $\log(n)$, so that the empirical slope can be compared with the convergence rates predicted by the theory. For real datasets, the true regression function is unavailable, and hence the goal is not to verify theorems directly; instead, these experiments illustrate the practical implications of the theory, especially the effect of the observation resolution or FPCA truncation level on prediction accuracy and the resulting trade-off in choosing $K_n$.

**Summary of Simulation Studies.** Due to space constraints and to maintain a clear presentation, we briefly summarize

the simulation results here and provide full details in Appendix D. Our experiments evaluated the performance of Linear Models (LM) and Deep Neural Networks (DNN) across six cases, ranging from targets driven by localized features to those based on global aggregates. There are two main observations. First, in terms of model flexibility, DNNs consistently approximate the underlying regression function across all cases, whereas LMs perform well in linear settings but degrade substantially in nonlinear ones. Second, we investigated how the dimensionality of the input representation affects DNN learning efficiency, using both binning and FPCA as dimensionality reduction methods. Although FPCA typically provides more efficient representations than binning at comparable dimensions due to its parsimony, the performance for both approaches reveals a clear trade-off between resolution and learnability. We further analyze the empirical error decay of DNNs and find an approximately linear relationship with sample size on a log-log scale, matching the theoretical results in Section 3.

## 4.1. Data Application

We apply our analysis to two real-world datasets: household electricity consumption and physical activity monitoring. Through these applications, we empirically investigate how different functional data representations influence the predictive performance of DNNs in practical scalar-on-function regression and classification tasks.

**Electricity Data.** We evaluate our framework on a smart-meter dataset collected from 5,098 London households (https://data.london.gov.uk/dataset/smartmeter-energy-consumption-data-in-london-households-vqm0d/). The objective is to predict future peak energy consumption based on historical data. The functional covariate $X_i$ is constructed from half-hourly energy readings recorded during the winter period of November 5, 2012, to February 28, 2013. This period was selected to capture heating-driven consumption patterns, as daily high temperatures during this time remained below 60°F. To mitigate the volatility of raw time series, we compute an average weekly profile for each household using only weekdays. This process yields a smooth 240-dimensional functional vector representing the typical weekday diurnal cycle. Figure 1 visualizes the resulting average profiles for two randomly selected households. For the response variable $Y_i$, we consider two tasks: a regression task predicting the maximum energy consumption observed in the subsequent testing window of January 2014, and a classification task predicting whether this peak usage exceeds a data-driven threshold.

**Wearable Device Data**. We analyze the relationship between daily physical activity patterns and depression severity using data from the NHANES 2013-2014 cycle (https:

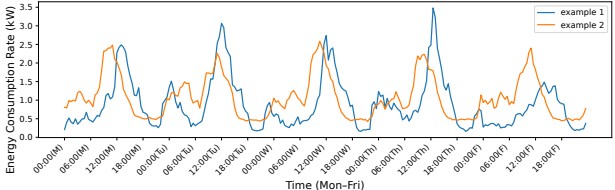

*Figure 1.* Two randomly selected functional covariate examples from the UK electricity dataset.

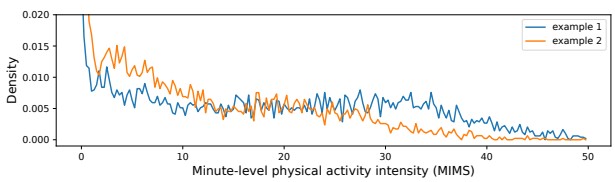

*Figure 2.* Two randomly selected functional covariate examples from the NHANES dataset.

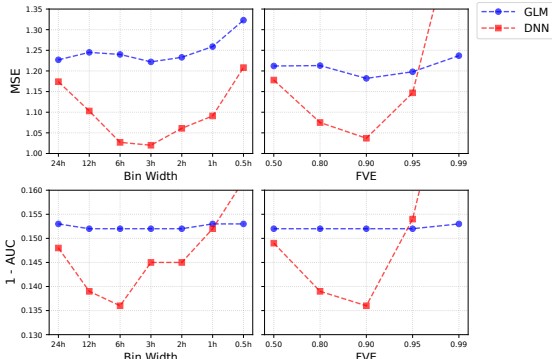

*Figure 3.* **Predictive performance on the UK Household Electricity dataset.** The top row displays MSE versus bin width and MSE versus FVE; the bottom row displays $1 - \text{AUC}$ versus the number of bins and $1 - \text{AUC}$ versus FVE.

//wwwn.cdc.gov/nchs/nhanes/continuousn
hanes/default.aspx?BeginYear=2013). The link between activity and mental health is of particular clinical interest among those with sleep issues. Therefore, we focused our analysis on adults who self-identified as "poor sleepers." This group includes individuals reporting less than 6 hours of sleep per night, as well as those with diagnosed sleep disorders or chronic trouble sleeping. The final analytical cohort consisted of $N = 1{,}623$ subjects. The functional covariate $X_i$ is derived from wrist-worn accelerometer data measuring Monitor-Independent Movement Summary units. Unlike the electricity data which captures temporal patterns, here we construct $X_i$ as the probability density function of activity intensity, pooled from five complete weekdays and discretized into 200 bins. This distribution captures how intensely a person moves rather than just when they move. The response $Y_i$ is based on the total score from the Patient Health Questionnaire-9 (PHQ-9). We evaluate performance on a regression task predicting this score and a classification task identifying moderate depression.

**Input Preprocessing Strategies.** We compared two strategies for reducing the dimensionality of $X_i$:

*Binning.* The dense input dimension is reduced by computing local averages within $d$ non-overlapping, equal-width bins. We evaluated a range of resolutions for each dataset: $d \in \{5, 10, 20, 50, 100, 200\}$ for the NHANES data and $d \in \{5, 10, 20, 40, 80, 120, 240\}$ for the UK data. For the UK data, the chosen values of $d$ correspond to averaging windows of 24, 12, 6, 3, 2, 1, and 0.5 hours, respectively.

*Functional Principal Components (FPC).* The functional covariates were represented by their first $K$ FPC scores, calculated from $X_i$. The number of components $K$ was determined by Fraction-of-Variance-Explained (FVE) of

0.5, 0.8, 0.9, 0.95, and 0.99.

**Models.** We compared two modeling approaches:

*Generalized Linear Models (GLM).* We used linear regression for regression tasks and logistic regression with a logit link function for classification tasks.

*Deep Neural Networks (DNN).* We implemented a multilayer perceptron in PyTorch. The architecture consisted of two hidden layers with 32 units each and ReLU activation functions. Models were trained using the Adam optimizer with a learning rate of $3 \times 10^{-3}$ and a batch size of $2^8$. Training ran for a maximum of 1,000 epochs, with early stopping triggered if validation loss failed to improve for 500 epochs. For the classification tasks, the network was trained using the negative log-likelihood loss.

**Experimental Setup.** For both applications, we evaluated performance using random subsampling. After randomly shuffling the data, 20% of the samples were held out as a fixed test set. The remaining data were split into training and validation sets in a 3:1 ratio. This procedure was repeated 200 times, and we report the average mean squared error (MSE) and classification error $(1-\text{AUC})$ over these runs, all evaluated on the same test set.

**Results.** The results for both the regression and classification tasks exhibit a consistent pattern regarding the dimensionality of the functional input, validating our hypothesis that a balance between resolution and efficiency is critical for capturing the nuanced functional features required for prediction, such as predicting peak energy consumption in the UK electricity data or estimating PHQ-9 depression scores in the NHANES data. As shown in Figures 3 and 4, we observe a distinct U-shaped error curve for the DNN models across both preprocessing methods. Performance is suboptimal at very low dimensions, where aggressive smoothing eliminates the high-frequency local information

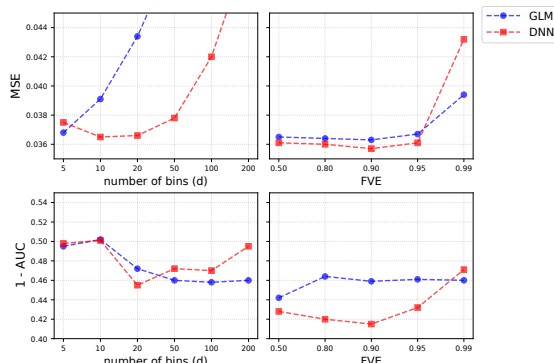

*Figure 4.* **Predictive performance on the NHANES Physical Activity dataset.** The panels follow the same order as Figure 3.

necessary to identify these critical features. Similarly, performance degrades at very high dimensions, likely due to overfitting to noise and the "curse of dimensionality". Consequently, optimal performance is consistently achieved at intermediate resolutions: specifically, using 3- to 6-hour bin widths for the UK dataset and 20 bins for the NHANES dataset, or an FPCA representation explaining 90% of the variance. This demonstrates that while dimensionality reduction is beneficial, retaining sufficient granularity is essential when the target variable depends on fine-grained functional characteristics rather than global averages.

## Impact Statement

This paper provides the first convergence-rate analysis for deep neural network regression with functional covariates. Beyond theoretical guarantees, our results clarify how finite-dimensional projections of functional inputs affect learning. These insights extend to other settings involving function-valued data, such as time-series modeling and operator learning. As functional data may encode sensitive attributes, we encourage responsible downstream use with appropriate privacy, robustness, and fairness considerations.

## Acknowledgements

The computing resources for this work were supported in part by the University of North Carolina at Chapel Hill through its AI Acceleration Program. HZ is partially supported by the SDSS Computational Data Science & AI Cloud Pilot Program at the University of North Carolina at Chapel Hill. XCD is partially supported by NSF DMS-2515104. JLW is partially supported by NSF DMS-24-13924.

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

# A. When Can Depth Improve the Convergence Rate?

In functional linear and functional generalized linear models, the finite-dimensional target after FPCA truncation is essentially linear in the principal component scores. Therefore, additional network depth is not needed for approximation: a shallow ReLU network can already represent affine maps exactly. This does not mean that depth is irrelevant for functional regression. Rather, depth becomes beneficial when the regression function has a hierarchical compositional structure after being projected onto a small number of functional directions.

We illustrate this point through a structured nonlinear scalar-on-function model. Let $Y_i = f_\star(X_i) + \varepsilon_i$ where

$$f_\star(X) = g_\star\big(\eta_1(X), \ldots, \eta_q(X)\big), \tag{10}$$

with $\eta_j(X) = \int_0^1 \{X(t) - \mu(t)\}\theta_j(t)\, dt$. Write $\theta_j(t) = \sum_{k=1}^\infty \theta_{jk}\phi_k(t)$, $|\theta_{jk}| \le C_\theta k^{-b}$, $j = 1, \ldots, q$, for some $b > 1/2$. For $K \ge 1$, define the truncated functional indices $\eta_{j,K}(\boldsymbol{\xi}) = \sum_{k=1}^K \theta_{jk}\xi_k$, $\boldsymbol{\eta}_K(\boldsymbol{\xi}) = \{\eta_{1,K}(\boldsymbol{\xi}), \ldots, \eta_{q,K}(\boldsymbol{\xi})\}^\top$. Thus $g_\star\{\boldsymbol{\eta}_K(\boldsymbol{\xi})\}$ is the finite-dimensional approximation to $f_\star(X)$ based on the first $K$ principal component scores.

The key structural assumption is that $g_\star$ is not an arbitrary $q$-dimensional nonparametric function, but instead has a hierarchical compositional form. Let $\mathcal{G}_{\mathrm{comp}}(\alpha, d_{\max}, Q, C)$ denote a class of functions that can be written as a composition of $Q$ layers, $g_\star = g_Q \circ g_{Q-1} \circ \cdots \circ g_0$, where each coordinate function in each layer depends on at most $d_{\max}$ arguments, has Hölder smoothness $\alpha$, and has Hölder norm bounded by $C$. We assume throughout this subsection that $g_\star$ is bounded and Lipschitz. The important case is $d_{\max} \ll q$, for example a binary-tree interaction structure with $d_{\max} = 2$. The following proposition shows that, under this structure, a deep ReLU network can achieve a rate depending on $d_{\max}$, the local dimension of the composition, rather than the ambient index dimension $q$.

**Proposition A.1.** *Assume Assumptions 3.1 and 3.2. Consider model (10) with $g_\star \in \mathcal{G}_{\mathrm{comp}}(\alpha, d_{\max}, Q, C)$, where $g_\star$ is bounded and Lipschitz. Let $\hat{f}$ be the empirical risk minimizer in (6) over a ReLU network class $\mathcal{F}_n$ with input dimension $K_n$, depth at least $cQ$ for a constant $c > 0$, uniformly bounded output $\sup_{f \in \mathcal{F}_n} \|f\|_\infty \le M_{\mathcal{F}}$, and total number of parameters $P_n$. Suppose that the covering number of $\mathcal{F}_n$ satisfies, for $\delta \in (0, 1)$,*

$$\log \mathcal{N}(\mathcal{F}_n, \|\cdot\|_\infty, \delta) \lesssim (qK_n + P_n) \log\left(\frac{n}{\delta}\right).$$

*Then, up to logarithmic factors,*

$$\mathfrak{R}(\hat{f}, f_\star) = O_P\left\{K_n^{1-a-2b} + P_n^{-\frac{2\alpha}{d_{\max}}} + \frac{qK_n + P_n}{n}\right\}.$$

*Consequently, choosing $K_n \asymp n^{1/(a+2b)}$, $P_n \asymp n^{d_{\max}/(2\alpha+d_{\max})}$, gives*

$$\mathfrak{R}(\hat{f}, f_\star) = \widetilde{O}_P\left\{n^{-\frac{a+2b-1}{a+2b}} + n^{-\frac{2\alpha}{2\alpha+d_{\max}}}\right\},$$

*where $\widetilde{O}_P(\cdot)$ hides logarithmic factors.*

If the hierarchical structure of $g_\star$ is ignored and $g_\star$ is treated as a generic $q$-dimensional Hölder function, then the corresponding approximation term for a shallow network is of order $P_n^{-2\alpha/q}$. Balancing this term with the estimation term gives the slower nonparametric rate $n^{-2\alpha/(2\alpha+q)}$. By contrast, the deep-network rate in Proposition A.1 depends on $d_{\max}$, the largest local dimension of the component functions, rather than on the ambient index dimension $q$. Thus, when $d_{\max} \ll q$, depth leads to a strictly faster rate. For example, if $q = 2^Q$ and $d_{\max} = 2$, so a network with depth proportional to $Q$ attains the nonlinear rate $n^{-2\alpha/(2\alpha+2)}$ up to logarithmic factors, whereas a shallow architecture that approximates the same function as an unstructured $q$-variate function yields $n^{-2\alpha/(2\alpha+q)}$. This explains why the linear benchmark studied earlier does not require depth, while genuinely hierarchical nonlinear functional regression models can benefit substantially from deeper architectures.

# B. Proofs of Theorems

*Proof of Theorem 3.4.* Under the functional linear model (1) and Assumption 3.3, the regression function satisfies

$$f(X) = \mathbb{E}(Y \mid X) = \int_0^1 \{X(t) - \mu(t)\}\beta(t)\, dt = \sum_{k=1}^\infty \xi_k b_k.$$

Let $K := K_n$ and define $f_0(\boldsymbol{\xi}) = \sum_{k=1}^{K} \xi_k b_k$, and $r_K(X) = \sum_{k=K+1}^{\infty} \xi_k b_k$, so that $f(X) = f_0(\boldsymbol{\xi}) + r_K(X)$. For an independent copy $X^*$ with scores $\boldsymbol{\xi}^*$, by $(a+b)^2 \le 2a^2 + 2b^2$,

$$\mathfrak{R}(\hat{f}, f) = \mathbb{E}_* \left[ \{f_0(\boldsymbol{\xi}^*) - \hat{f}(\boldsymbol{\xi}^*) + r_K(X^*)\}^2 \right] \le 2 \mathbb{E}_* \left[ \{f_0(\boldsymbol{\xi}^*) - \hat{f}(\boldsymbol{\xi}^*)\}^2 \right] + 2 \mathbb{E}_* \left[ r_K(X^*)^2 \right]. \tag{11}$$

Since the principal component scores are uncorrelated and $\mathbb{E}[(\xi_k^*)^2] = \lambda_k$,

$$\mathbb{E}_* \left[ r_K(X^*)^2 \right] = \mathbb{E}_* \left[ \left( \sum_{k=K+1}^{\infty} \xi_k^* b_k \right)^2 \right] = \sum_{k=K+1}^{\infty} b_k^2 \lambda_k \lesssim \sum_{k=K+1}^{\infty} k^{-a-2b} \lesssim K^{1-a-2b}. \tag{12}$$

Let $T_n = C \log n$ with a constant $C > 0$ that is chosen later, and define $\tilde{f}_0(\boldsymbol{\xi}) = \sum_{k=1}^{K} \xi_k \mathbf{1}_{\{|\xi_k| \le T_n\}} b_k$. Then

$$\mathbb{E} \left[ \{f_0(\boldsymbol{\xi}^*) - \hat{f}(\boldsymbol{\xi}^*)\}^2 \right] \le 2 \mathbb{E} \left[ \{\tilde{f}_0(\boldsymbol{\xi}^*) - \hat{f}(\boldsymbol{\xi}^*)\}^2 \right] + 2 \mathbb{E} \left[ \{f_0(\boldsymbol{\xi}^*) - \tilde{f}_0(\boldsymbol{\xi}^*)\}^2 \right]. \tag{13}$$

Moreover, $f_0(\boldsymbol{\xi}^*) - \tilde{f}_0(\boldsymbol{\xi}^*) = \sum_{k=1}^{K} \xi_k^* \mathbf{1}_{\{|\xi_k^*| > T_n\}} b_k$. Using $(\sum_{k=1}^{K} a_k)^2 \le K \sum_{k=1}^{K} a_k^2$ and Cauchy–Schwarz,

$$\mathbb{E} \left[ \{f_0(\boldsymbol{\xi}^*) - \tilde{f}_0(\boldsymbol{\xi}^*)\}^2 \right] \le K \sum_{k=1}^{K} b_k^2 \mathbb{E} \left[ (\xi_k^*)^2 \mathbf{1}_{\{|\xi_k^*| > T_n\}} \right] \le K \sum_{k=1}^{K} b_k^2 \left( \mathbb{E}[(\xi_k^*)^4] \right)^{1/2} \mathbb{P}(|\xi_k^*| > T_n)^{1/2}.$$

By Assumption 3.2, the $\xi_k$'s are uniformly sub-exponential for $k \le K$, hence $\sup_{k \le K} \mathbb{E}[(\xi_k^*)^4] < \infty$ and $\sup_{k \le K} \mathbb{P}(|\xi_k^*| > T_n) \lesssim e^{-cT_n}$ for some $c > 0$. Since $\sum_{k \ge 1} b_k^2 < \infty$,

$$\mathbb{E} \left[ \{f_0(\boldsymbol{\xi}^*) - \tilde{f}_0(\boldsymbol{\xi}^*)\}^2 \right] \lesssim K \, e^{-cT_n} = K \, n^{-cC} = o(n^{-1})$$

for $C$ large enough.

Now write $\mathfrak{R}(\hat{f}, f_0) := \mathbb{E}_*[\{f_0(\boldsymbol{\xi}^*) - \hat{f}(\boldsymbol{\xi}^*)\}^2]$. The following lemma extends the bias-variance decomposition of the prediction error $\mathfrak{R}(\hat{f}, \tilde{f}_0)$ under the functional linear model (1). Similar types of bias-variance decompositions for regression with fixed-dimensional multivariate covariates over compact domains have been previously studied, for example, in Lemma 4 of Schmidt-Hieber (2020) and Proposition 4 of Suzuki (2019). Its proof can be found in Appendix C.

**Lemma B.1.** *Let $P$ denote the law of $\boldsymbol{\xi} = (\xi_1, \ldots, \xi_{K_n})^\top$ and let $\mathfrak{R}(\hat{f}, f_0) := \mathbb{E}_*[\{f_0(\boldsymbol{\xi}^*) - \hat{f}(\boldsymbol{\xi}^*)\}^2]$, where $\boldsymbol{\xi}^* \sim P$ is independent of the training sample. Assume model (1) and that there exists a measurable set $\mathcal{X} \subset \mathbb{R}^{K_n}$ such that $\boldsymbol{\xi} \in \mathcal{X}$ almost surely. Assume also that $\sup_{f \in \mathcal{F}} \|f\|_\infty \le M_\mathcal{F}$, where $\| \cdot \|_\infty$ is the supremum norm over $\mathcal{X}$. Let $\hat{f}$ be the ERM in (6), and for $\delta > 0$ let $N_\delta := \mathcal{N}(\mathcal{F}, \| \cdot \|_\infty, \delta)$ with $\log N_\delta \ge 1$. Then for any $\delta > 0$,*

$$\mathbb{E} \left[ \mathfrak{R}(\hat{f}, f_0) \right] \lesssim \inf_{f \in \mathcal{F}} \|f - f_0\|_{L_P^2}^2 + M_\mathcal{F} \delta + \frac{(M_\mathcal{F}^2 + B_n^2 + \sigma^2) \log N_\delta}{n} + \frac{M_\mathcal{F}}{\sqrt{n}} K_n^{\frac{1-a-2b}{2}},$$

*where $\sigma^2 := \mathbb{E}[\varepsilon^2]$ and $B_n$ is any deterministic sequence satisfying $\mathbb{P}(\max_{i \le n} |f_0(\boldsymbol{\xi}_i)| > B_n) \le n^{-2}$.*

In particular, since $\mathcal{F}$ contains linear maps (e.g. $x \mapsto x$ can be represented by $\sigma(x) - \sigma(-x)$), it contains $\boldsymbol{\xi} \mapsto \sum_{k=1}^{K} b_k \xi_k$, hence $\inf_{f \in \mathcal{F}} \|f - f_0\|_{\mathcal{L}_P^2}^2 = 0$. Using the covering-number bound $\log N_\delta \lesssim K \log (K \log n / \delta)$ and choosing $\delta = n^{-1}$, Lemma B.1 implies

$$\mathbb{E} \left[ \mathfrak{R}(\hat{f}, f_0) \right] \lesssim \frac{(M_\mathcal{F}^2 + B_n^2) K \log n}{n} + \frac{M_\mathcal{F}}{\sqrt{n}} K^{\frac{1-a-2b}{2}} + o(n^{-1}).$$

Because $\mathfrak{R}(\hat{f}, f_0) \ge 0$, Markov's inequality implies the corresponding probability bound

$$\mathfrak{R}(\hat{f}, f_0) = O_P \left( \frac{(M_\mathcal{F}^2 + B_n^2) K \log n}{n} + \frac{M_\mathcal{F}}{\sqrt{n}} K^{\frac{1-a-2b}{2}} \right). \tag{14}$$

The second term in (14) is of smaller order than the first at the choice of $K$ below, so it can be absorbed into the $O_P(\cdot)$ term.

Combining (11), (12), and (14) yields

$$\mathfrak{R}(\hat{f}, f) \lesssim \frac{(M_\mathcal{F}^2 + B_n^2) K \log n}{n} + K^{1-a-2b} + o(n^{-1}).$$

Choosing $K = K_n \asymp n^{1/(a+2b)}$ (ignoring logarithmic factors) gives

$$\mathfrak{R}(\hat{f}, f) = O_P\left(n^{\frac{1-a-2b}{a+2b}} (M_\mathcal{F}^2 + B_n^2) \log n\right).$$

In particular, if $f_0(\boldsymbol{\xi})$ is sub-Gaussian then one may take $B_n^2 \asymp \log n$, yielding an additional $\log n$ factor; if only sub-exponential tails are available, one typically takes $B_n \asymp \log n$, yielding an additional $(\log n)^2$ factor. Regarding the order of $M_\mathcal{F}$, one can work on the truncated compact event $\mathcal{X}_n = \{\|\boldsymbol{\xi}\|_\infty \le \log n\}$, and, by a similar argument, it can be shown that $\mathcal{X}_n$ occurs with high probability. $\square$

*Proof of Theorem 3.7.* Let $K := K_n$ and write $f(X) = \sum_{k=1}^\infty \xi_k b_k$, $f_0(\boldsymbol{\xi}) = \sum_{k=1}^K \xi_k b_k$, $\boldsymbol{\xi} = (\xi_1, \dots, \xi_K)^\top$. For an independent test point $X^*$ with scores $\boldsymbol{\xi}^*$, use the three-point identity for Bregman divergences: for any $u, v, w$,

$$\mathcal{B}_\psi(u, w) = \mathcal{B}_\psi(u, v) + \mathcal{B}_\psi(v, w) + \{\psi'(v) - \psi'(w)\}(u - v).$$

Taking $u = \hat{f}(\boldsymbol{\xi}^*)$, $v = f_0(\boldsymbol{\xi}^*)$, and $w = f(X^*)$ gives

$$\mathcal{B}_\psi(\hat{f}(\boldsymbol{\xi}^*), f(X^*)) = \mathcal{B}_\psi(\hat{f}(\boldsymbol{\xi}^*), f_0(\boldsymbol{\xi}^*)) + \mathcal{B}_\psi(f_0(\boldsymbol{\xi}^*), f(X^*)) + \Delta^*, \tag{15}$$

where $\Delta^* := \{\psi'(f_0(\boldsymbol{\xi}^*)) - \psi'(f(X^*))\}\{\hat{f}(\boldsymbol{\xi}^*) - f_0(\boldsymbol{\xi}^*)\}$.

Since $Y \mid X \sim Q_{f(X)}$ is a one-parameter exponential family, $\mathcal{B}_\psi(u, v) = \mathrm{KL}(Q_v \,\|\, Q_u)$ for natural parameters $u, v$. The change-of-measure argument of Dou et al. (2012) implies that under Assumptions 3.1, 3.5, and 3.6,

$$\mathbb{E}_*\left[\mathcal{B}_\psi(f_0(\boldsymbol{\xi}^*), f(X^*))\right] \lesssim \mathbb{E}_*\left[\{f_0(\boldsymbol{\xi}^*) - f(X^*)\}^2\right]. \tag{16}$$

Moreover, by uncorrelatedness of scores and $\mathbb{E}[(\xi_k^*)^2] = \lambda_k$, $\mathbb{E}_*\left[\{f_0(\boldsymbol{\xi}^*) - f(X^*)\}^2\right] = \sum_{k>K} b_k^2 \lambda_k \lesssim \sum_{k>K} k^{-(a+2b)} \lesssim K^{1-a-2b}$. Hence

$$\mathbb{E}_*\left[\mathcal{B}_\psi(f_0(\boldsymbol{\xi}^*), f(X^*))\right] \lesssim K^{1-a-2b}. \tag{17}$$

Let $B_n := C\sqrt{\log n}$ for a sufficiently large constant $C > 0$ and define the event $\mathcal{E}_n := \{|f_0(\boldsymbol{\xi}^*)| \le B_n\}$. Since $f_0(\boldsymbol{\xi}^*)$ is sub-Gaussian with $\mathbb{E}[f_0(\boldsymbol{\xi}^*)^2] \asymp 1$, we have $\mathbb{P}(\mathcal{E}_n^c) \le n^{-A}$ for any fixed $A > 0$ by choosing $C$ large enough.

On $\mathcal{E}_n$, because $\|\hat{f}\|_\infty \le M_\mathcal{F}$, Taylor's expansion implies that for some $c^*$ between $\hat{f}(\boldsymbol{\xi}^*)$ and $f_0(\boldsymbol{\xi}^*)$, $\mathcal{B}_\psi(\hat{f}(\boldsymbol{\xi}^*), f_0(\boldsymbol{\xi}^*)) = \frac{1}{2}\psi''(c^*)\{\hat{f}(\boldsymbol{\xi}^*) - f_0(\boldsymbol{\xi}^*)\}^2$. Let $m_n := \inf_{|t| \le M_\mathcal{F} \vee B_n} \psi''(t) > 0$. Then on $\mathcal{E}_n$,

$$\{\hat{f}(\boldsymbol{\xi}^*) - f_0(\boldsymbol{\xi}^*)\}^2 \mathbf{1}_{\mathcal{E}_n} \le \frac{2}{m_n} \mathcal{B}_\psi(\hat{f}(\boldsymbol{\xi}^*), f_0(\boldsymbol{\xi}^*)) \mathbf{1}_{\mathcal{E}_n}. \tag{18}$$

By Cauchy–Schwarz,

$$\mathbb{E}_*[|\Delta^*|\mathbf{1}_{\mathcal{E}_n}] \le \left(\mathbb{E}_*[\{\psi'(f_0(\boldsymbol{\xi}^*)) - \psi'(f(X^*))\}^2]\right)^{1/2} \left(\mathbb{E}_*[\{\hat{f}(\boldsymbol{\xi}^*) - f_0(\boldsymbol{\xi}^*)\}^2 \mathbf{1}_{\mathcal{E}_n}]\right)^{1/2}. \tag{19}$$

By the mean-value theorem, $\psi'(f_0) - \psi'(f) = \psi''(\tilde{f})(f_0 - f)$ for some $\tilde{f}$ between $f_0$ and $f$. The same change-of-measure control as in Dou et al. (2012) implies $\mathbb{E}_*\left[\{\psi'(f_0(\boldsymbol{\xi}^*)) - \psi'(f(X^*))\}^2\right]^{1/2} \lesssim \mathbb{E}_*[(f_0(\boldsymbol{\xi}^*) - f(X^*))^2]^{1/2} \lesssim K^{\frac{1-a-2b}{2}}$. Combining this with (18) gives

$$\mathbb{E}_*[|\Delta^*|\mathbf{1}_{\mathcal{E}_n}] \lesssim \sqrt{\frac{1}{m_n}} K^{\frac{1-a-2b}{2}} \left(\mathbb{E}_*[\mathcal{B}_\psi(\hat{f}(\boldsymbol{\xi}^*), f_0(\boldsymbol{\xi}^*))]\right)^{1/2}.$$

Applying AM–GM yields

$$\mathbb{E}_*[|\Delta^*|\mathbf{1}_{\mathcal{E}_n}] \lesssim \mathbb{E}_*[\mathcal{B}_\psi(\hat{f}(\boldsymbol{\xi}^*), f_0(\boldsymbol{\xi}^*))] + \frac{1}{m_n} K^{1-a-2b}. \tag{20}$$

On $\mathcal{E}_n^c$ we use the crude bound $|\hat{f} - f_0| \leq M_{\mathcal{F}} + |f_0|$ and Cauchy–Schwarz to obtain $\mathbb{E}_*[|\Delta^*| \mathbf{1}_{\mathcal{E}_n^c}] = o(K^{1-a-2b})$ by the choice of $B_n$. Therefore,

$$\mathbb{E}_*[|\Delta^*|] \lesssim \mathbb{E}_*[\mathcal{B}_\psi(\hat{f}(\boldsymbol{\xi}^*), f_0(\boldsymbol{\xi}^*))] + \frac{1}{m_n} K^{1-a-2b}. \tag{21}$$

Define

$$\ell_n(f) := \frac{1}{n} \sum_{i=1}^n \{-Y_i f(\boldsymbol{\xi}_i) + \psi(f(\boldsymbol{\xi}_i))\}, \quad \hat{f} \in \arg\min_{f \in \mathcal{F}} \ell_n(f),$$

and $R_0(f) := \mathbb{E}_*[\mathcal{B}_\psi(f(\boldsymbol{\xi}^*), f_0(\boldsymbol{\xi}^*))]$. By a similar argument as in the proof of Lemma B.1, one has the standard generalization bound for any $\delta > 0$ with $\log N_\delta \geq 1$, where $N_\delta := \mathcal{N}(\mathcal{F}, \|\cdot\|_\infty, \delta)$,

$$\mathbb{E}[R_0(\hat{f})] \lesssim \inf_{f \in \mathcal{F}} R_0(f) + M_{\mathcal{F}} \delta + \frac{(M_{\mathcal{F}}^2 + B_n^2) \log N_\delta}{n}. \tag{22}$$

Since $\mathcal{F}$ contains linear maps, it can exactly represent $f_0(\boldsymbol{\xi}) = \sum_{k \leq K} b_k \xi_k$, hence $\inf_{f \in \mathcal{F}} R_0(f) = 0$. Take $\delta = n^{-1}$ and use $\log N_\delta \lesssim K \log(\delta^{-1} K \log n)$ to get $\mathbb{E}[R_0(\hat{f})] \lesssim K(\log n)^2/n$. Since $R_0(\hat{f}) \geq 0$, Markov's inequality implies

$$R_0(\hat{f}) = O_P\left(\frac{K(\log n)^2}{n}\right). \tag{23}$$

Taking $\mathbb{E}_*$ in (15) and combining (17), (21) and (23) yields

$$\mathbb{E}_*\left[\mathcal{B}_\psi(\hat{f}(\boldsymbol{\xi}^*), f(X^*))\right] \lesssim R_0(\hat{f}) + \frac{1}{m_n} K^{1-a-2b} = O_P\left(\frac{K(\log n)^2}{n} + K^{1-a-2b}\right),$$

where $m_n^{-1}$ is absorbed into logarithmic factors under Assumption 3.6. Choosing $K = K_n \asymp n^{1/(a+2b)}$ (up to log factors) gives

$$\mathbb{E}_*\left[\mathcal{B}_\psi(\hat{f}(\boldsymbol{\xi}^*), f(X^*))\right] = O_P\left(n^{\frac{1-a-2b}{a+2b}} (\log n)^2\right),$$

which completes the proof. $\square$

*Proof of Theorem 3.8.* We prove the lower bound via Le Cam's two-point method. Since $X$ is centered Gaussian with eigenvalues $\{\lambda_k\}$, we may write $\|X\|^2 = S := \sum_{k=1}^\infty \lambda_k Z_k^2$, where $\{Z_k\}_{k \geq 1}$ are i.i.d. standard normal random variables. For any $t > 0$, by Markov's inequality,

$$\mathbb{P}(S \leq h^2) = \mathbb{P}\left(e^{-tS} \geq e^{-th^2}\right) \leq e^{th^2} \mathbb{E}\left[e^{-tS}\right] = e^{th^2} \prod_{k=1}^\infty (1 + 2t\lambda_k)^{-1/2},$$

where we used $\mathbb{E}[e^{-t\lambda Z^2}] = (1 + 2t\lambda)^{-1/2}$ for $Z \sim \mathcal{N}(0, 1)$. Hence

$$\log \mathbb{P}(\|X\| \leq h) \leq th^2 - \frac{1}{2} \sum_{k=1}^\infty \log(1 + 2t\lambda_k). \tag{24}$$

By Assumption 3.1, there exist constants $c_0 > 0$ and $k_0 \in \mathbb{N}$ such that $\lambda_k \geq c_0 k^{-a}$ for all $k \geq k_0$. Thus for $t$ large,

$$\sum_{k=1}^\infty \log(1 + 2t\lambda_k) \geq \sum_{k=k_0}^\infty \log\left(1 + c_1 t k^{-a}\right) \geq \int_{k_0}^\infty \log\left(1 + c_1 t x^{-a}\right) \, dx,$$

where $c_1 := 2c_0$ and we used that $x \mapsto \log(1 + c_1 t x^{-a})$ is decreasing. With the change of variables $x = (c_1 t)^{1/a} y$,

$$\int_{k_0}^\infty \log\left(1 + c_1 t x^{-a}\right) \, dx = (c_1 t)^{1/a} \int_{k_0(c_1 t)^{-1/a}}^\infty \log\left(1 + y^{-a}\right) \, dy \geq c_2 t^{1/a}$$

for some constant $c_2 > 0$ and all sufficiently large $t$, because $\int_0^\infty \log(1 + y^{-a}) \, dy \in (0, \infty)$ when $a > 1$. Plugging into (24) yields $\log \mathbb{P}(\|X\| \leq h) \leq th^2 - c_3 t^{1/a}$ for all sufficiently large $t$, for some constant $c_3 > 0$. Choosing $t = \kappa h^{-2a/(a-1)}$

with $\kappa > 0$ small enough gives $\mathbb{P}\left(\|X\| \le h\right) \le \exp\left\{-c_4 h^{-2/(a-1)}\right\}$ for all sufficiently small $h$, and some constant $c_4 > 0$.

Fix $h > 0$ and define $f_0(x) \equiv 0$, $f_1(x) := L\left(h^p - \|x\|^p\right)_+$. Since $p \in (0,1]$, the map $r \mapsto r^p$ is $p$-Hölder on $\mathbb{R}^+$, and by the triangle inequality $|\|x\| - \|y\|| \le \|x - y\|$, hence $|\|x\|^p - \|y\|^p| \le |\|x\| - \|y\||^p \le \|x - y\|^p$. Because $u \mapsto (h^p - u)_+$ is 1-Lipschitz, it follows that $|f_1(x) - f_1(y)| \le L\|x - y\|^p$, so $f_0, f_1 \in \mathfrak{H}(L,p)$. Moreover, $f_0(\mathsf{o}) = 0$, $f_1(\mathsf{o}) = Lh^p =: \Delta$.

Let $\mathbb{P}_j^{(n)}$ be the joint law of $\{(X_i, Y_i)\}_{i=1}^n$ under $f = f_j$. A standard two-point argument implies that for any estimator $\hat{f}_n(\mathsf{o})$,

$$\max_{j \in \{0,1\}} \mathbb{E}_{f_j}\left[(\hat{f}_n(\mathsf{o}) - f_j(\mathsf{o}))^2\right] \ge \frac{\Delta^2}{8}\left(1 - \|\mathbb{P}_0^{(n)} - \mathbb{P}_1^{(n)}\|_{\mathrm{TV}}\right). \tag{25}$$

By Pinsker's inequality, $\|\mathbb{P}_0^{(n)} - \mathbb{P}_1^{(n)}\|_{\mathrm{TV}}^2 \le \mathrm{KL}(\mathbb{P}_0^{(n)}\|\mathbb{P}_1^{(n)})/2$. Under Gaussian noise, conditional on $X_i$ we have $Y_i \mid X_i \sim \mathcal{N}(f_j(X_i), \sigma^2)$, hence

$$\mathrm{KL}\left(\mathbb{P}_0^{(n)} \,\|\, \mathbb{P}_1^{(n)}\right) = \frac{n}{2\sigma^2}\,\mathbb{E}\left[\{f_1(X) - f_0(X)\}^2\right] = \frac{n}{2\sigma^2}\,\mathbb{E}\left[f_1(X)^2\right].$$

Since $|f_1(X)| \le Lh^p$ and $f_1(X) = 0$ whenever $\|X\| > h$, $\mathbb{E}\left[f_1(X)^2\right] \le L^2 h^{2p}\,\mathbb{P}\left(\|X\| \le h\right)$, and therefore

$$\mathrm{KL}(\mathbb{P}_0^{(n)}\|\mathbb{P}_1^{(n)}) \le \frac{nL^2 h^{2p}}{2\sigma^2}\mathbb{P}(\|X\| \le h). \tag{26}$$

Let $h_n := C(\log n)^{-(a-1)/2}$ with $C > 0$ chosen sufficiently small so that $c_4 C^{-2/(a-1)} \ge 4$. Then $\mathbb{P}(\|X\| \le h_n) \le \exp\{-c_4 h_n^{-2/(a-1)}\} = n^{-c_4 C^{-2/(a-1)}} \le n^{-4}$. Plugging into (26) yields

$$\mathrm{KL}\left(\mathbb{P}_0^{(n)} \,\|\, \mathbb{P}_1^{(n)}\right) \le \frac{nL^2 h_n^{2p}}{2\sigma^2}\, n^{-4} = O\left(\frac{(\log n)^{-p(a-1)}}{n^3}\right) \to 0.$$

Hence $\|\mathbb{P}_0^{(n)} - \mathbb{P}_1^{(n)}\|_{\mathrm{TV}} \to 0$, and (25) yields, for all large $n$,

$$\sup_{f \in \mathfrak{H}(L,p)} \mathbb{E}_f\left[\left\{\hat{f}_n(\mathsf{o}) - f(\mathsf{o})\right\}^2\right] \ge c\,\Delta^2 = c\,L^2 h_n^{2p} \asymp (\log n)^{-p(a-1)}.$$

Finally, since $n^\gamma(\log n)^{-p(a-1)} \to \infty$ for every $\gamma > 0$, no polynomial pointwise rate is achievable. $\qquad \square$

## C. Proofs of Auxiliary Results

*Proof of Lemma B.1.* Write $P$ for the law of $\boldsymbol{\xi}$ and $\mathbb{P}_n$ for the empirical measure on $\boldsymbol{\xi}_1, \ldots, \boldsymbol{\xi}_n$. Throughout, $\|\cdot\|_\infty$ denotes the supremum norm on a measurable set $\mathcal{X} \subset \mathbb{R}^{K_n}$ such that $\boldsymbol{\xi} \in \mathcal{X}$ almost surely, and we assume $\sup_{f \in \mathcal{F}} \|f\|_\infty \le M_{\mathcal{F}}$. Note that $\mathfrak{R}(\hat{f}, f_0) = \mathbb{P}\{(\hat{f} - f_0)^2\} = \|\hat{f} - f_0\|_{\mathcal{L}_P^2}^2$. Let $\boldsymbol{\xi}_1', \ldots, \boldsymbol{\xi}_n'$ be an i.i.d. sample from $P$, independent of $\{(\boldsymbol{\xi}_i, Y_i)\}_{i=1}^n$. Then

$$\mathbb{E}\left[\mathfrak{R}(\hat{f}, f_0)\right] = \mathbb{E}\left[\frac{1}{n}\sum_{i=1}^n \{\hat{f}(\boldsymbol{\xi}_i') - f_0(\boldsymbol{\xi}_i')\}^2\right].$$

Define

$$D := \left| \mathbb{E}\left[\frac{1}{n}\sum_{i=1}^n \left(\{\hat{f}(\boldsymbol{\xi}_i) - f_0(\boldsymbol{\xi}_i)\}^2 - \{\hat{f}(\boldsymbol{\xi}_i') - f_0(\boldsymbol{\xi}_i')\}^2\right)\right]\right|.$$

Then

$$\mathbb{E}\left[\mathfrak{R}(\hat{f}, f_0)\right] \le \mathbb{E}\left[\mathbb{P}_n\{(\hat{f} - f_0)^2\}\right] + D. \tag{27}$$

Let $\{f_1, \ldots, f_{N_\delta}\}$ be a $\delta$-net of $\mathcal{F}$ in $\|\cdot\|_\infty$, where $N_\delta = \mathcal{N}(\mathcal{F}, \|\cdot\|_\infty, \delta)$. For each realization of $\hat{f}$, choose an index $J$ such that $\|\hat{f} - f_J\|_\infty \le \delta$. For any $\boldsymbol{\xi} \in \mathcal{X}$,

$$\{\hat{f}(\boldsymbol{\xi}) - f_0(\boldsymbol{\xi})\}^2 - \{f_J(\boldsymbol{\xi}) - f_0(\boldsymbol{\xi})\}^2 = \{\hat{f}(\boldsymbol{\xi}) - f_J(\boldsymbol{\xi})\}\{\hat{f}(\boldsymbol{\xi}) + f_J(\boldsymbol{\xi}) - 2f_0(\boldsymbol{\xi})\},$$

hence by $\|\hat{f} - f_J\|_\infty \leq \delta$ and $|\hat{f}|, |f_J| \leq M_{\mathcal{F}}$,

$$\left| \{\hat{f}(\boldsymbol{\xi}) - f_0(\boldsymbol{\xi})\}^2 - \{f_J(\boldsymbol{\xi}) - f_0(\boldsymbol{\xi})\}^2 \right| \leq 2\delta \left( |\hat{f}(\boldsymbol{\xi})| + |f_J(\boldsymbol{\xi})| + 2|f_0(\boldsymbol{\xi})| \right) \leq 4\delta \left( M_{\mathcal{F}} + |f_0(\boldsymbol{\xi})| \right).$$

Applying this bound to both $\boldsymbol{\xi}_i$ and $\boldsymbol{\xi}'_i$ yields

$$D \leq \frac{1}{n}\mathbb{E}\left| \sum_{i=1}^n g_J(\boldsymbol{\xi}_i, \boldsymbol{\xi}'_i) \right| + C_1\, \delta\, (M_{\mathcal{F}} + \|f_0\|_{L_P^2}), \tag{28}$$

where $g_j(\boldsymbol{\xi}, \boldsymbol{\xi}') := \{f_j(\boldsymbol{\xi}) - f_0(\boldsymbol{\xi})\}^2 - \{f_j(\boldsymbol{\xi}') - f_0(\boldsymbol{\xi}')\}^2$, and $C_1 > 0$ is an absolute constant.

Fix $A > 0$ and define $r_j := \max\{A,\ \|f_j - f_0\|_{L_P^2}\}$. Let

$$T := \max_{1 \leq j \leq N_\delta} \left| \sum_{i=1}^n \frac{g_j(\boldsymbol{\xi}_i, \boldsymbol{\xi}'_i)}{r_j} \right|.$$

Then $|\sum_{i=1}^n g_J(\boldsymbol{\xi}_i, \boldsymbol{\xi}'_i)| \leq r_J T$, so by AM–GM,

$$\frac{1}{n}\mathbb{E}[r_J T] \leq \frac{1}{2}\mathbb{E}[r_J^2] + \frac{1}{2n^2}\mathbb{E}[T^2].$$

Moreover,

$$r_J^2 \leq A^2 + \|f_J - f_0\|_{L_P^2}^2 \leq A^2 + 2\|\hat{f} - f_0\|_{L_P^2}^2 + 2\|\hat{f} - f_J\|_\infty^2 \leq A^2 + 2\,\mathfrak{R}(\hat{f}, f_0) + 2\delta^2,$$

hence

$$\frac{1}{n}\mathbb{E}\left| \sum_{i=1}^n g_J(\boldsymbol{\xi}_i, \boldsymbol{\xi}'_i) \right| \leq \frac{1}{2}\mathbb{E}\left[ \mathfrak{R}(\hat{f}, f_0) \right] + \frac{1}{2}A^2 + \frac{1}{2n^2}\mathbb{E}[T^2] + C_2\delta^2, \tag{29}$$

for an absolute constant $C_2 > 0$.

Let $\mathfrak{B}_n := \{\max_{i \leq n} |f_0(\boldsymbol{\xi}_i)| \leq B_n,\ \max_{i \leq n} |f_0(\boldsymbol{\xi}'_i)| \leq B_n\}$. On $\mathfrak{B}_n$, we have $|f_j(\boldsymbol{\xi}) - f_0(\boldsymbol{\xi})| \leq M_{\mathcal{F}} + B_n$. In particular,

$$\{f_j(\boldsymbol{\xi}) - f_0(\boldsymbol{\xi})\}^4 \leq (M_{\mathcal{F}} + B_n)^2 \{f_j(\boldsymbol{\xi}) - f_0(\boldsymbol{\xi})\}^2,$$

and thus, using $(a - b)^2 \leq 2a^2 + 2b^2$ and independence of $\boldsymbol{\xi}, \boldsymbol{\xi}'$,

$$\mathbb{E}\left[ \left( \frac{g_j(\boldsymbol{\xi}, \boldsymbol{\xi}')}{r_j} \right)^2 \middle| \mathfrak{B}_n \right] \leq \frac{4}{r_j^2}\mathbb{E}\left[ \{f_j(\boldsymbol{\xi}) - f_0(\boldsymbol{\xi})\}^4 \middle| \mathfrak{B}_n \right] \leq 4(M_{\mathcal{F}} + B_n)^2,$$

since $r_j \geq \|f_j - f_0\|_{L_P^2}$. Also, $g_j(\boldsymbol{\xi}_i, \boldsymbol{\xi}'_i)$ is centered and independent across $i$ for fixed $j$, and $|g_j(\boldsymbol{\xi}, \boldsymbol{\xi}')/r_j| \leq 2(M_{\mathcal{F}} + B_n)^2/A$ because $r_j \geq A$. By Bernstein's inequality and a union bound over $j = 1, \ldots, N_\delta$, there exists $C_3 > 0$ such that for all $t > 0$,

$$\mathbb{P}\left(T > t,\ \mathfrak{B}_n\right) \leq 2N_\delta \exp\left( -\frac{C_3\, t^2}{n(M_{\mathcal{F}} + B_n)^2 + (M_{\mathcal{F}} + B_n)^2 t/A} \right).$$

Integrating this tail bound yields the standard estimate

$$\mathbb{E}[T^2 \mathbf{1}_{\mathfrak{B}_n}] \lesssim n(M_{\mathcal{F}} + B_n)^2 \log N_\delta \lesssim n(M_{\mathcal{F}}^2 + B_n^2) \log N_\delta. \tag{30}$$

On $\mathfrak{B}_n^c$ we use the crude bound $T^2 \leq C_4 n^2 \max_i (M_{\mathcal{F}} + |f_0(\boldsymbol{\xi}_i)|)^4$ and Cauchy–Schwarz to obtain $\mathbb{E}[T^2 \mathbf{1}_{\mathfrak{B}_n^c}] \leq \mathbb{E}[T^2]/2 + o\left(n(M_{\mathcal{F}}^2 + B_n^2) \log N_\delta\right)$ provided $B_n$ is chosen so that $\mathbb{P}(\mathfrak{B}_n^c) \leq n^{-2}$ and $f_0(\boldsymbol{\xi})$ has finite fourth moment. Consequently,

$$\mathbb{E}[T^2] \lesssim n(M_{\mathcal{F}}^2 + B_n^2) \log N_\delta. \tag{31}$$

Now set $A^2 := c_0(M_{\mathcal{F}}^2 + B_n^2) \log N_\delta / n$ with a sufficiently large absolute constant $c_0$. Combining (28), (29), and (31) gives

$$D \leq \frac{1}{2}\mathbb{E}\left[ \mathfrak{R}(\hat{f}, f_0) \right] + C\,\delta(M_{\mathcal{F}} + \|f_0\|_{L_P^2}) + C\,\frac{(M_{\mathcal{F}}^2 + B_n^2) \log N_\delta}{n}.$$

Plugging this into (27) and moving the $\mathbb{E}[\mathfrak{R}(\hat{f}, f_0)]/2$ term to the left yields

$$\mathbb{E}\left[\mathfrak{R}(\hat{f}, f_0)\right] \lesssim \mathbb{E}\left[\mathbb{P}_n\{(\hat{f} - f_0)^2\}\right] + M_{\mathcal{F}}\delta + \frac{(M_{\mathcal{F}}^2 + B_n^2)\log N_\delta}{n}. \tag{32}$$

Write $Y_i = f_0(\boldsymbol{\xi}_i) + u_i$ with $u_i := \varepsilon_i + r_i$ and $r_i := \sum_{k > K_n} \xi_{ik} b_k$. Since $\hat{f}$ minimizes the empirical squared loss, $\mathbb{P}_n(Y - \hat{f})^2 \leq \mathbb{P}_n(Y - f)^2$ for all $f \in \mathcal{F}$. Expanding $Y = f_0 + u$ gives

$$\mathbb{P}_n(\hat{f} - f_0)^2 \leq \mathbb{P}_n(f - f_0)^2 + \frac{2}{n}\sum_{i=1}^n u_i\{\hat{f}(\boldsymbol{\xi}_i) - f(\boldsymbol{\xi}_i)\}.$$

Taking expectation and minimizing over $f \in \mathcal{F}$ yields

$$\mathbb{E}\left[\mathbb{P}_n\{(\hat{f} - f_0)^2\}\right] \leq \inf_{f \in \mathcal{F}} \|f - f_0\|_{L_P^2}^2 + \frac{2}{n}\mathbb{E}\left[\sum_{i=1}^n \varepsilon_i \hat{f}(\boldsymbol{\xi}_i)\right] + \frac{2}{n}\mathbb{E}\left[\sum_{i=1}^n r_i \hat{f}(\boldsymbol{\xi}_i)\right]. \tag{33}$$

The noise term is handled by the standard covering argument as in Suzuki (2019): for any $\delta > 0$,

$$\left|\frac{2}{n}\mathbb{E}\left[\sum_{i=1}^n \varepsilon_i \hat{f}(\boldsymbol{\xi}_i)\right]\right| \leq \frac{1}{2}\mathbb{E}\left[\mathbb{P}_n\{(\hat{f} - f_0)^2\}\right] + C\left(M_{\mathcal{F}}\delta + \frac{\sigma^2(\log N_\delta + 1)}{n}\right), \tag{34}$$

where $\sigma^2 := \mathbb{E}[\varepsilon^2]$. For the tail term, by Cauchy–Schwarz and $|\hat{f}| \leq M_{\mathcal{F}}$,

$$\left|\frac{2}{n}\mathbb{E}\left[\sum_{i=1}^n r_i \hat{f}(\boldsymbol{\xi}_i)\right]\right| \leq \frac{2}{n}\left(\mathbb{E}\left[\left(\sum_{i=1}^n r_i\right)^2\right]\right)^{1/2} \left(\mathbb{E}[\hat{f}(\boldsymbol{\xi})^2]\right)^{1/2} \leq \frac{2M_{\mathcal{F}}}{n}\sqrt{n\,\mathbb{E}[r_1^2]} \lesssim \frac{M_{\mathcal{F}}}{\sqrt{n}}\,K_n^{\frac{1-a-2b}{2}},$$

since $\mathbb{E}[r_1^2] = \sum_{k > K_n} \lambda_k b_k^2 \lesssim K_n^{1-a-2b}$. Combining (33)–(34) yields

$$\mathbb{E}\left[\mathbb{P}_n\{(\hat{f} - f_0)^2\}\right] \lesssim \inf_{f \in \mathcal{F}} \|f - f_0\|_{L_P^2}^2 + M_{\mathcal{F}}\delta + \frac{\sigma^2 \log N_\delta}{n} + \frac{M_{\mathcal{F}}}{\sqrt{n}}\,K_n^{\frac{1-a-2b}{2}}. \tag{35}$$

Substituting (35) into (32) yields the claimed bound. Finally, if $f_0(\boldsymbol{\xi})$ is sub-Gaussian, choose $B_n \asymp \sqrt{\log n}$ so that $\mathbb{P}(\mathfrak{B}_n^c) \leq n^{-2}$; then $B_n^2 \asymp \log n$ and the third term becomes $\{(M_{\mathcal{F}}^2 + \log n)\log N_\delta\}/n$. $\qquad\square$

*Proof of Proposition A.1.* Let $m_K(\boldsymbol{\xi}) := \mathbb{E}\{f_\star(X) \mid \xi_1, \ldots, \xi_K\}$ be the population regression function based on the first $K$ principal component scores. Then $Y_i = m_K(\boldsymbol{\xi}_i) + \zeta_i$, $\mathbb{E}(\zeta_i \mid \boldsymbol{\xi}_i) = 0$.

We first bound the FPCA truncation error. Let $h_K(\boldsymbol{\xi}) := g_\star\{\boldsymbol{\eta}_K(\boldsymbol{\xi})\}$. Since $m_K$ is the $L^2$-projection of $f_\star(X)$ onto the sigma-field generated by $(\xi_1, \ldots, \xi_K)$, $\mathbb{E}\{f_\star(X) - m_K(\boldsymbol{\xi})\}^2 \leq \mathbb{E}\{f_\star(X) - h_K(\boldsymbol{\xi})\}^2$. Because $g_\star$ is Lipschitz,

$$\mathbb{E}\{f_\star(X) - h_K(\boldsymbol{\xi})\}^2 \lesssim \sum_{j=1}^q \mathbb{E}\{\eta_j(X) - \eta_{j,K}(\boldsymbol{\xi})\}^2 = \sum_{j=1}^q \sum_{k > K} \lambda_k \theta_{jk}^2 \lesssim \sum_{k > K} k^{-a-2b} \lesssim K^{1-a-2b}.$$

Therefore,

$$\mathbb{E}\{f_\star(X) - m_K(\boldsymbol{\xi})\}^2 \lesssim K^{1-a-2b}. \tag{36}$$

Next, we bound the approximation error of $m_K$ by a deep ReLU network. By the standard ReLU approximation result for compositional Hölder functions, since $g_\star \in \mathcal{G}_{\text{comp}}(\alpha, d_{\max}, Q, C)$, there exists a ReLU network $\widetilde{g}$ with depth proportional to $Q$ and $P_n$ parameters such that $\sup_{\boldsymbol{u}} |\widetilde{g}(\boldsymbol{u}) - g_\star(\boldsymbol{u})| \lesssim P_n^{-\alpha/d_{\max}}$, where the supremum is over the relevant truncated domain of the functional indices. The first affine layer of the network can compute the $q$ truncated indices $\eta_{j,K}(\boldsymbol{\xi}) = \sum_{k=1}^K \theta_{jk} \xi_k$, $j = 1, \ldots, q$, using $qK$ parameters. Hence the composed map $\widetilde{f}(\boldsymbol{\xi}) := \widetilde{g}\{\boldsymbol{\eta}_K(\boldsymbol{\xi})\}$ belongs to a ReLU network class with depth at least $cQ$ and parameter count of order $qK + P_n$. Moreover, $\|\widetilde{f} - h_K\|_{L_P^2}^2 \lesssim P_n^{-2\alpha/d_{\max}}$. Using the projection identity again,

$$\|h_K - m_K\|_{L_P^2}^2 \leq \mathbb{E}\{h_K(\boldsymbol{\xi}) - f_\star(X)\}^2 \lesssim K^{1-a-2b}.$$

Therefore,

$$\inf_{f \in \mathcal{F}_n} \|f - m_K\|_{L_P^2}^2 \lesssim P_n^{-2\alpha/d_{\max}} + K^{1-a-2b}. \tag{37}$$

It remains to control the statistical estimation error. Applying the same ghost-sample and covering-number argument as in Lemma B.1, now with $m_K$ replacing $f_0$, yields, for any $\delta > 0$,

$$\mathbb{E}\|\hat{f} - m_K\|_{L_P^2}^2 \lesssim \inf_{f \in \mathcal{F}_n} \|f - m_K\|_{L_P^2}^2 + M_{\mathcal{F}}\delta + \frac{(M_{\mathcal{F}}^2 + B_n^2 + \sigma^2)\log\mathcal{N}(\mathcal{F}_n, \|\cdot\|_\infty, \delta)}{n},$$

where $B_n$ is a deterministic truncation level satisfying the same high-probability condition as in Lemma B.1. Taking $\delta = n^{-1}$ and using the assumed covering bound gives

$$\|\hat{f} - m_K\|_{L_P^2}^2 = O_P\left\{P_n^{-2\alpha/d_{\max}} + K^{1-a-2b} + \frac{(qK + P_n)\log n}{n}\right\},$$

up to the same logarithmic factors coming from $B_n$.

Finally, by the orthogonal decomposition of prediction error,

$$\mathfrak{R}(\hat{f}, f_\star) = \mathbb{E}_*\{\hat{f}(\boldsymbol{\xi}^*) - m_K(\boldsymbol{\xi}^*)\}^2 + \mathbb{E}_*\{m_K(\boldsymbol{\xi}^*) - f_\star(X^*)\}^2.$$

Combining this identity with (36) and (37) completes the proof. $\square$

# D. Numerical Experiments

We conducted a comprehensive set of simulations to evaluate the predictive performance of different functional data analysis approaches. The experiments were designed to test the models under various underlying regression structures, ranging from simple linear interactions to complex nonlinear dependencies.

## D.1. Data Generating Processes and Evaluation Metrics

**Functional Predictor Generation.** The functional covariate $X(t)$ was defined on the domain $t \in [0, 1]$. We generated each trajectory as a linear combination of $K = 50$ cosine basis functions, $\phi_k(t) = \sqrt{2}\cos(2\pi kt)$: $X(t) = \sum_{k=1}^K \xi_k \phi_k(t)$, where the basis coefficients $\xi_k$ were independently sampled from a normal distribution with decaying variance, $\xi_k \sim \mathcal{N}(0, k^{-2})$. Each function $X(t)$ was evaluated on a dense equispaced grid of 200 points in $[0, 1]$ to form the dense vector $X_{\text{dense}}$. $X_{\text{dense}}$ is assumed to be measurement error free.

**Response Generation Scenarios.** The true regression function $f(X)$ was defined differently across six cases. Let $\beta(t) = t - 0.3$:

1. **Case 1 (Single Linear Score):** $f(X) = \xi_3$.

2. **Case 2 (Single Quadratic Score):** $f(X) = \xi_3^2$.

3. **Case 3 (Interaction):** $f(X) = \left(\int_0^{0.7} X(t)\mathrm{d}t\right) \times \left(\int_{0.7}^1 X(t)\mathrm{d}t\right)$.

4. **Case 4 (Linear):** $f(X) = \int_0^1 X(t)\beta(t)\mathrm{d}t$.

5. **Case 5 (Quadratic Functional):** $f(X) = \left(\int_0^1 X(t)\beta(t)\mathrm{d}t\right)^2$.

6. **Case 6 (Logistic):** $f(X) = \int_0^1 X(t)\beta(t)\mathrm{d}t$, determining the outcome via $\mathrm{logit}[P(Y = 1 \mid X)] = f(X)$.

Cases 1 through 5 are regression tasks; the scalar response was generated as $Y = f(X) + \epsilon$, with noise $\epsilon \sim \mathcal{N}(0, 0.1)$. For Case 6, the binary outcome $Y \in \{0, 1\}$ was generated according to the specified logistic probability, which is a standard classification task.

**Input Preprocessing Strategies.** We compared two strategies for reducing the dimensionality of $X_{\text{dense}}$ prior to modeling:

*Binning.* The dense input was reduced to dimension $d$ by averaging values within $d$ non-overlapping, equal-width bins. We denote the resulting predictor $X_{\text{bin}} \in \mathbb{R}^d$ and varied the number of bins (NoB) $d \in \{1, 5, 10, 20, 50, 100\}$.

*Functional Principal Components (FPC).* The functional covariates were represented by their first $K$ FPC scores, calculated directly from $X_{\text{dense}}$. The number of components $K$ was determined by Fraction-of-Variance-Explained (FVE) thresholds of 0.9, 0.95, and 0.99, corresponding to $K = 3$, $K = 6$, and $K = 19$, respectively.

**Models.** We fitted the preprocessed inputs using two types of models:

*Generalized Linear Models (GLM).* We used linear regression for continuous responses and logistic regression with a logit link function for the binary classification case.

*Deep Neural Networks (DNN).* We implemented a multi-layer perceptron in PyTorch. The architecture consisted of 2 hidden layers with 32 units each and ReLU activation functions. Models were trained using the Adam optimizer with a learning rate of $3 \times 10^{-3}$ and a batch size of 256. Training ran for a maximum of 1000 epochs, with early stopping triggered if validation loss failed to improve for 500 epochs. For Case 6, the network was trained using the negative log-likelihood loss.

**Evaluation Metrics and Procedure.** Model performance was evaluated using the prediction error risk. For models $\widehat{f}$ using FPC scores, we computed $\mathfrak{R}(\hat{f}, f)$, consistent with the theoretical rate established in Theorem 3.4. For models $\widehat{f}$ using binned inputs, we evaluated the analogous risk $\mathbb{E}_*[\{f(X^*) - \hat{f}(X^*_{\text{bin}})\}^2]$, where $X^*_{\text{bin}}$ is the binned representation of the independent test copy $X^*$. This metric serves as a direct parallel to the PC-based risk, effectively replacing the projection onto eigenbasis functions with projection onto the binning basis functions. The expectation was estimated on an independently generated universal test set of 1,000 samples for each case. We varied the training sample size $n \in \{2^{11}, 2^{12}, 2^{13}, 2^{14}, 2^{15}\}$. For every configuration, 200 independent datasets were generated, and the reported results represent the average risk across these replications. We visualize performance by plotting the log-risk against $\log(n)$ to examine convergence rates.

## D.2. Results

*Performance of LM.* The efficacy of the linear model is strictly determined by the alignment between the model assumption and the true regression structure. In scenarios where the true link function is linear (Cases 1, 4, and 6), the LM demonstrates superior efficiency, consistently outperforming DNNs (Figures 5, 8, and 10). This is expected, as the LM is the correctly specified model for these cases. Notably, for the global integration tasks (Cases 4 and 6), LMs maintain robust performance even with low-dimensional representations (small number of bins or low FVE), as the target function relies on global aggregation rather than fine-grained local details (Figures 8 and 10). As dimensionality increases (e.g., using 100 bins or high FVE), we observe a slight drop in efficiency due to redundant parameters. Conversely, in nonlinear scenarios (Cases 2, 3, and 5), the LM fails to capture the underlying relationships, resulting in substantial bias that does not vanish with increasing sample size (Figures 6, 7, and 9).

*Performance of DNN.* DNNs demonstrate their universal approximation capabilities by achieving consistent convergence across all six cases, regardless of linearity. However, the optimal dimensionality reduction strategy for DNNs heavily depends on whether the target function is "global" or "local." In Cases 4 to 6 where the response $Y$ depends on global information, lower-dimensional representations generally suffice (Figures 8, 9, and 10). The DNNs perform efficiently with fewer bins or PC scores, as the high-frequency details discarded by dimensionality reduction are irrelevant to the global integral. In contrast, for cases driven by specific local features, we observe a distinct "sweet spot" in dimensionality. For instance, in Cases 1 and 2, where the response depends explicitly on the third score ($\xi_3$), the input representation must be rich enough to resolve this specific feature (Figures 5 and 6). We summarize the specific impact of this dimensionality constraint on both reduction strategies below:

- **Impact of Binning:** From Figures 5a and 6a, we find that intermediate binning resolutions (e.g., $d \approx 20$) yield optimal performance. When $d$ is too small, the bins are too coarse to capture the necessary local variations. When $d$ is too large (e.g., $d = 100$), the model suffers from the "curse of dimensionality," where the excess noise and parameters degrade learning efficiency.

- **Impact of FPCA:** A similar phenomenon is observed with FPC scores based on Figures 5b and 6b. Using a low FVE threshold (e.g., 0.5 or 0.8) retains only the first 1 to 2 components, discarding the critical third component ($\xi_3$) and making the model incapable of learning the target. Optimal performance is achieved only when the FVE threshold is

sufficiently high (0.9) to include $\xi_3$. However, increasing the threshold further (0.95 or 0.99) introduces higher-order noise components, which again causes a performance drop.

*Hybrid Dependencies.* Case 3, which involves an interaction between two disjoint sub-intervals ($\int_0^{0.7}$ and $\int_{0.7}^1$), represents a mixture of global aggregation and local partitioning. In Figure 7, we observe that while coarser representations (e.g., 5 bins or 1 PC) perform adequately at small sample sizes, they eventually saturate. For the binned input with $d = 5$, the error curve flattens as sample size increases, indicating a bias floor where the coarse bins cannot perfectly resolve the boundary at $t = 0.7$. In contrast, models with higher-dimensional inputs ($d = 20$ or higher FVE) maintain a steeper downward slope in error, eventually surpassing the coarser models as $n$ grows large ($n = 2^{15}$). This highlights that while aggressive dimensionality reduction is efficient for small datasets, finer resolutions are necessary to fully exploit the information in large-sample regimes.

*Convergence Rates.* Finally, across all successful model-case combinations, the plots of log-risk versus log-sample size ($\log \mathfrak{R}(\hat{f}, f)$ vs. $\log n$) exhibit clear linear trends. This empirical power-law decay aligns with our theoretical expectations and supports the validity of the convergence rates established in Theorem 3.4.

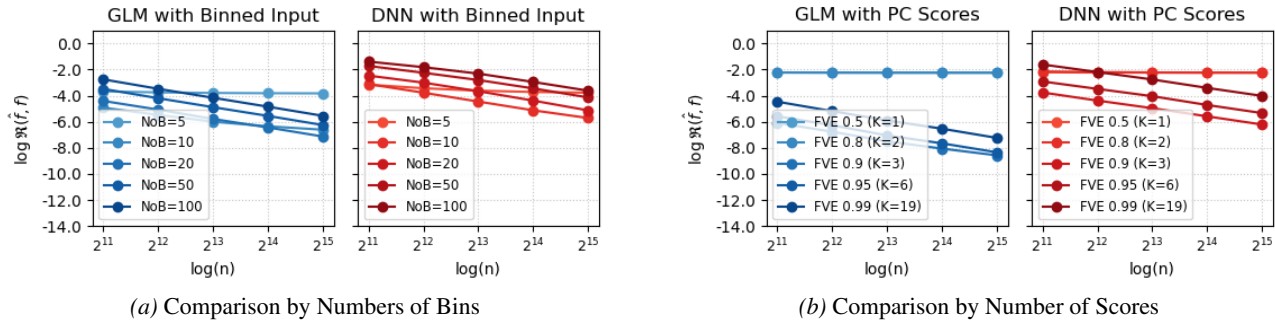

*(a) Comparison by Numbers of Bins*                    *(b) Comparison by Number of Scores*

*Figure 5.* Case 1. $\log \mathfrak{R}(\hat{f}, f)$ vs. $\log(n)$ – Summary of Method Comparisons

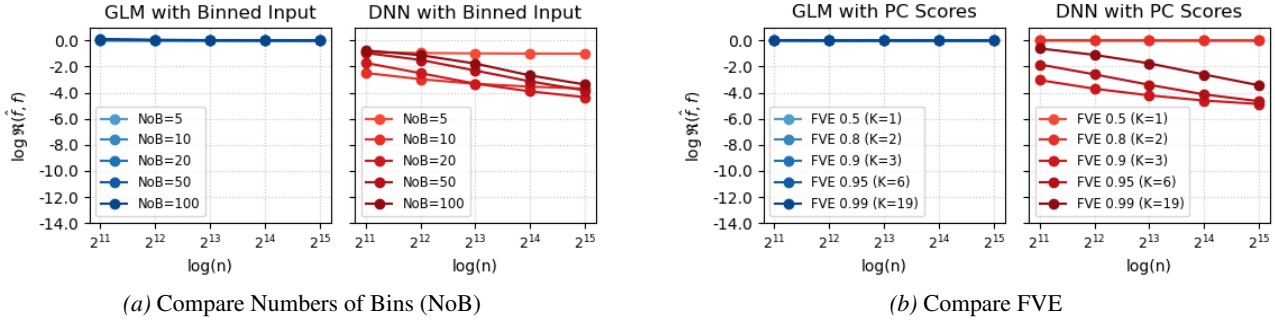

*(a) Compare Numbers of Bins (NoB)*                    *(b) Compare FVE*

*Figure 6.* Case 2. $\log \mathfrak{R}(\hat{f}, f)$ vs. $\log(n)$ – Summary of Method Comparisons

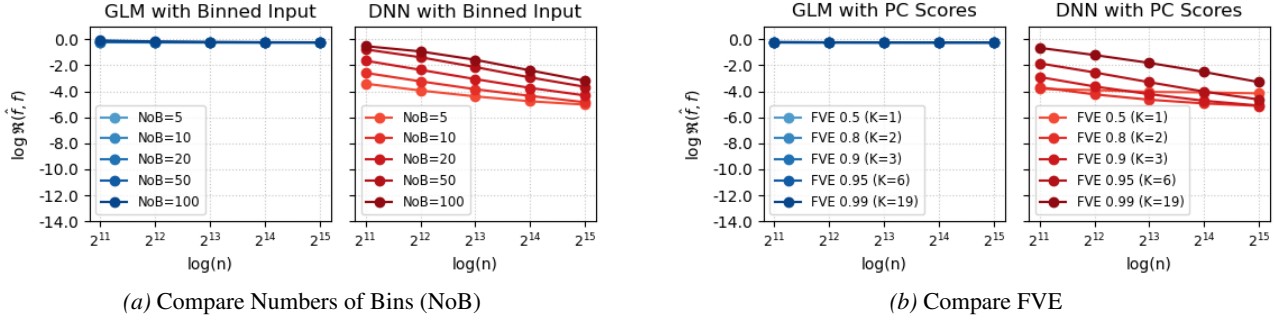

*(a) Compare Numbers of Bins (NoB)*                    *(b) Compare FVE*

*Figure 7.* Case 3. $\log \mathfrak{R}(\hat{f}, f)$ vs. $\log(n)$ – Summary of Method Comparisons

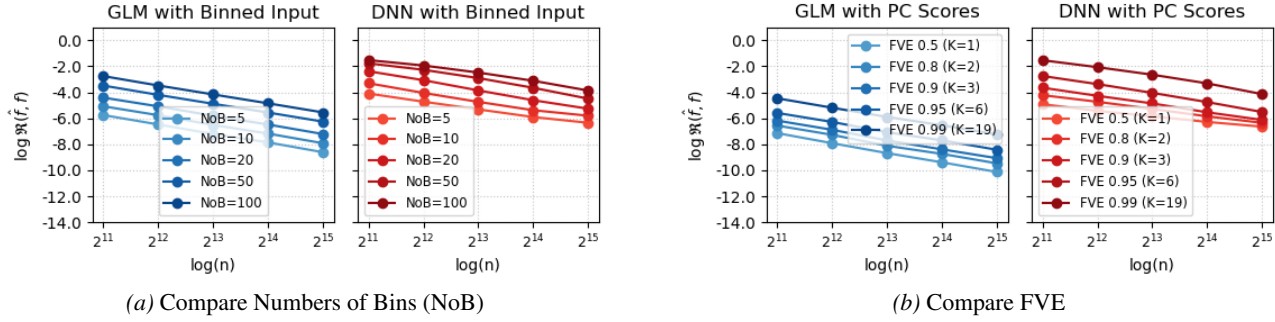

*(a)* Compare Numbers of Bins (NoB)          *(b)* Compare FVE

*Figure 8.* Case 4. $\log \Re(\hat{f}, f)$ vs. $\log(n)$ – Summary of Method Comparisons

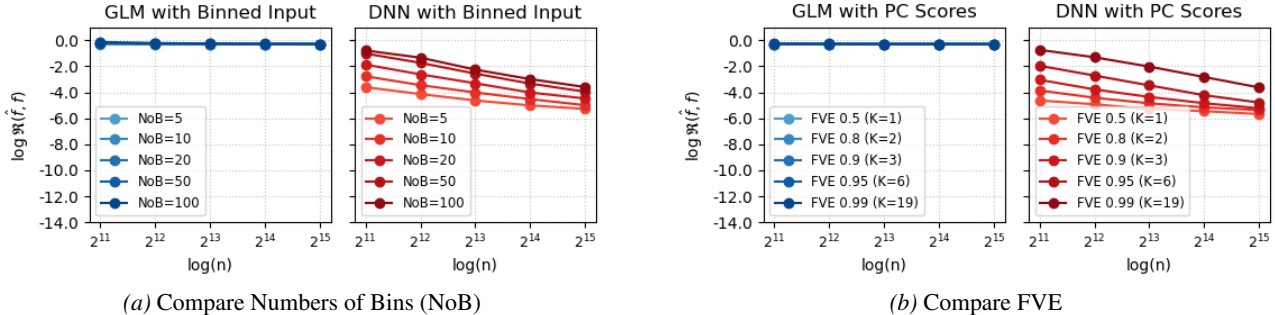

*(a)* Compare Numbers of Bins (NoB)          *(b)* Compare FVE

*Figure 9.* Case 5. $\log \Re(\hat{f}, f)$ vs. $\log(n)$ – Summary of Method Comparisons

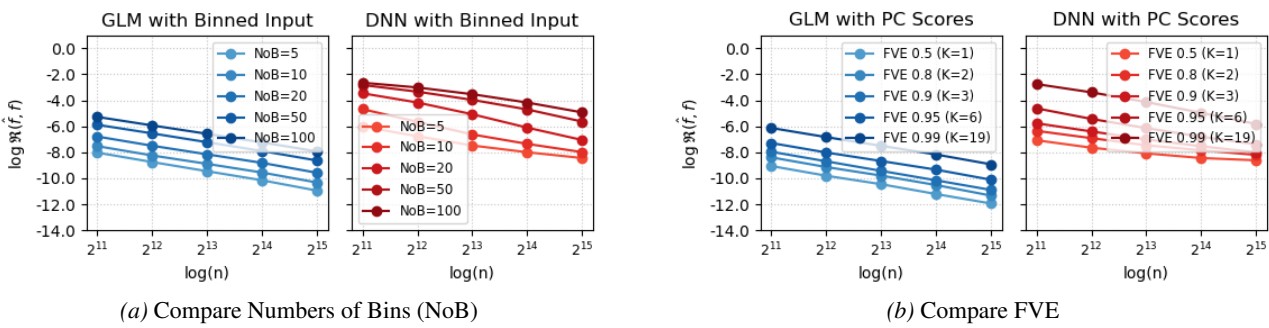

*(a)* Compare Numbers of Bins (NoB)          *(b)* Compare FVE

*Figure 10.* Case 6. $\log \Re(\hat{f}, f)$ vs. $\log(n)$ – Summary of Method Comparisons

