# OpenReview forum: "Deep Neural Network Regression with Functional Covariates"
_ICML.cc/2026/Conference — ICML 2026 regular_

### Official Review · Reviewer_Bw23 · 2026-03-04

**Soundness:** 3
**Presentation:** 3
**Significance:** 3
**Originality:** 3
**Overall Recommendation:** 3
**Confidence:** 4

**Summary:**

This paper studies the prediction error of regression with functional covariates with the deep neural network.

*Model elements*:
- The regression task takes the form $\min_{f \in \mathcal{F}} \frac{1}{n}\sum_{i=1}^n \|Y_i - f(X_i)\|^2$.
- The covariate (predictor) $X = X(t), t \in [0,1]$ is assumed to be in a function space $L^2$, admitting the Karhunen-Loeve expansion $X(t) = \mu(t) + \sum_{k=0}^{\infty} \xi_{k} \phi_k(t)$. When collecting the sample of $X$, $(\xi_{k}, k = 0,...,\infty)$ should be viewed as a sequence of random variables.
- The deep neural network is designed as a $L_D$-layer fully connected neural network (Equation (3)).
- The authors consider three scenarios of the underlying model of $Y | X$: (1) linear model: $Y = \int (X - \mu)(t) \beta(t) d t + \epsilon$; (2) nonlinear model with known link function: $Y | X \sim Q_\lambda, \lambda = \int X \beta$ and $Q_\lambda$ is in a known exponential family; (3) fully non parametric regression, where $Y = f(X) + \epsilon$ with $f$ being $L$-Lipschtiz.

**Contribution**:
- The authors show that in scenario (1) (2), under mild regularity conditions, the standard estimators (like least square estimator, maximum-likelihood estimator) attain the same optimal polynomial rates of convergence (up to a logarithmic factor) compared to a classic paper in the literature [1]. Especially in scenario (2), as the authors argue, this is a new result in the context of nonlinear regression.
- The authors provide a lower bound for scenario (3) when $X$ follows a Gaussian process, implying, without additional structural
assumptions, *polynomial* convergence rates generally cannot be achieved under standard conditions.

[1] Cai, T. T. and Hall, P. Prediction in functional linear regression. Annals of Statistics, 34(5):2159–2179, 2006.

**Compliance With Llm Reviewing Policy:**

Affirmed.

**Key Questions For Authors:**

1. How does $L_D$ impact the prediction error?

**Limitations:**

See the weaknesses above.

**Strengths And Weaknesses:**

> **Strength**:
- The writing is clear, and the literature study is comprehensive.
- The problem to address is significanly important in the sense that the theoretical result of regression using deep neural network under a wide range of models are long-standing target in the related field.

- The theoretical analysis (though it seems to be similar to the idea in [1]) shows the impact of dimensionality of the covaraite $X$ on the learning behavior: even though $X$ lives in an infinite dimensional space, then the expansion terms of $X$ in $X(t) = \mu(t) + \sum_{k=0}^{\infty} \xi_{k} \phi_k(t)$ decay fast enough (exponentially), then after suitable truncation on the series $(\xi_{k}, k = 0,...,\infty)$ to $(\xi_{k}, k = 0,...,K_n)$, and a careful bias-variance tradeoff in the error analysis to choose the truncation threshold $K_n$ ($n$ is the sample size), a polynomial rate of prediction error is achievable.

> **Weakness**:
In my opinions, the development of the main theory has the following *non-ignorable issues*:

1. The main theoretical insight that authors highlight in their analysis is the impact of dimensionality of $X$ on the learning rate (see above). However, the connection to the deep neural network is missing. Concretely, for example, the depth $L_D$ of the deep neural network never appears in the statement of the main theorems (e.g. Thm 3.4); in the proof of Thm 3.4 (page 11, line 600), the authors use the covering-number $N_\delta \leq O(K \log(K \log (L / \delta)))$, which is unjustified. (I guess the $L_D$ should come in through the covering number).

2. In scenario (1) (2), the task is basically to estimate a linear function $\xi \mapsto \sum_{k=1}^K b_k \xi_k$ (page 11, line 600), then given this structure, it is unclear why a deep neural network would be necessary, as the target function is linear and could be estimated directly using much simpler models. I understand that the result shows that the deep neural network can achieve an optimal polynomial rate (compared to [1]) under milder assumption, but the motivation of developing the prediction errors in these setting is unclear for me.

3. The result of scenario (3) gives a slow rate like $1/\log(n)$ without further exploration on more positive results, though the authors discuss more in-depth result in the literature. The discussion of scenario (3) is helpful but does not provide too much contribution in theory i think.

---

> ### Author Rebuttal · Authors · 2026-03-28
>
> Dear Reviewer Bw23,
>
> Thank you for the thoughtful and constructive review. We appreciate your recognition of the clarity, importance, and technical depth of the paper. We have revised the manuscript accordingly.
>
> Before our point-by-point responses, we clarify the relation to [1]. Both our method and [1] use FPCA truncation to regularize the infinite-dimensional covariate. The key difference is what happens after truncation. In [1], the estimator comes from the normal equations of a linear model, and the analysis relies on perturbation theory for compact operators. In contrast, our estimator is an empirical risk minimizer over a neural-network-based function class. This is more flexible and extends naturally beyond the linear model, in particular to functional GLMs. At the same time, our analysis still yields sharp rates depending on the smoothness of both the covariate process and the slope function.
>
> > The connection to the DNN is unclear; the depth $L_D$ does not appear in the theorems, and the covering-number argument seems unjustified.
>
> We agree that the entropy step is currently under-specified. In fact, the proof of Theorem 3.4 does not require the covering number of the full unrestricted DNN class. After truncating the functional predictor to the first $K_n$ FPC scores, the target in the functional linear and functional GLM settings becomes a $K_n$-dimensional linear predictor, which is already exactly representable by a ReLU network of depth $L_D=2$. Therefore, the leading convergence rate is governed by the effective input dimension $K_n$, not by the complexity of the full neural network class. The covering-number and empirical-process argument in Lemma A.1 is used primarily to obtain a generalization bound suited to the functional-data setting, where the predictor domain is unbounded and the input dimension increases with the sample size. When this linear subclass is viewed as embedded in a larger depth-$L_D$ ReLU family, the dependence on $L_D$ appears only through the ambient entropy bound and hence contributes at most logarithmic factors, without changing the leading polynomial exponent. We will clarify this explicitly in the revision.
>
> > In scenarios (1) and (2), the target is linear, so why use a deep neural network rather than a simpler model?
>
> We agree that if the target is known to be linear, simpler estimators are natural and often preferable in practice. Our goal is not to claim that DNNs are necessary for estimating $\sum_{k=1}^K b_k\xi_k$. Rather, we study a statistical learning question specific to functional data: how does a standard NN ERM behave when the input is infinite-dimensional? Even in the linear case, this is nontrivial because the representation involves infinitely many scores, the eigenvalues decay to zero, the effective dimension $K_n$ must grow with $n$, and the score distribution is typically noncompact.
>
> Scenarios (1) and (2) therefore serve as benchmark models where the optimal FDA rates are known, allowing us to isolate the effect of infinite-dimensional covariates on NN learning. Our results show that a standard ReLU-network ERM still attains the classical polynomial prediction rates, up to logarithmic factors, under milder assumptions. This benchmark analysis is also a first step toward understanding more flexible settings such as functional GLMs and fully nonparametric regression.
>
> > Scenario (3) only gives a slow rate like $1/\log n$ and seems less contributive.
>
> Scenario (3) is different in nature from scenarios (1) and (2) and is included for a complementary purpose. While scenarios (1) and (2) are benchmark settings where polynomial rates are achievable, scenario (3) characterizes the boundary of what is possible in fully nonparametric functional regression. It shows that without additional structural assumptions, polynomial rates are generally unattainable. The slow rate should therefore be interpreted not as a weakness of DNNs or of our proof technique, but as an intrinsic statistical limitation of the problem. We will revise the paper to make this benchmark-versus-boundary perspective explicit and to emphasize that scenario (3) motivates additional structure, such as single-index, sparse, or compositional assumptions, if one seeks polynomial rates.
>
> > How does $L_D$ impact the prediction error?
>
> In scenarios (1) and (2), once the functional covariate is truncated to the first $K_n$ scores, the target is a finite-dimensional linear predictor and can already be represented by a shallow ReLU network. Accordingly, depth does not affect the leading polynomial rate. Over a broader depth-$L_D$ ReLU class, $L_D$ enters only through the entropy bound and therefore only through logarithmic factors.
>
> [1] Cai, T. T., & Hall, P. (2006). Prediction in functional linear regression. Annals of Statistics, 34(5), 2159-2179.

---

> > ### Author Rebuttal · Reviewer_Bw23 · 2026-04-02
> >
> > 1. If from a theoretical point, the ReLU network of depth $L_D = 2$ is sufficient for the attained prediction error, then it seems that it does not explain the empirical success of the *deep* neural network as argued in the abstract.
> >
> > 2. In section 4, the author only consider a DNN of only 2 hidden layers, why not consider a deeper network?
> >
> > In summary, I think given the Weakness 1,3, the theory of this paper on explaining the prediction power of the DNN is a bit weak.

---

> > > ### Author Response · Authors · 2026-04-02
> > >
> > > Dear Reviewer Bw23,
> > >
> > > Thank you very much for your valuable feedback and follow-up questions. We sincerely appreciate your time and your recognition of our responses.
> > >
> > > We fully appreciate your concern about how depth affects the convergence rate and under what conditions a genuinely deep neural network can help. We also value your suggestion, and we are preparing examples in which the regression function has a more complex structure, such as compositions of smooth functions or generalized functional linear regression models with hierarchical parameter structures. In such cases, shallow neural networks may be insufficient, and deeper neural networks can be beneficial. Due to the time and space constraints of the rebuttal period, we may not be able to provide a complete discussion of these more complex settings here. However, in the revision we will add more details on scenarios in which deep neural networks truly help in regression settings with functional covariates.
> > >
> > > We provide more details regarding the theoretical contribution in the following point-by-point response to your questions.
> > >
> > > > If from a theoretical point, the ReLU network of depth $L_D=2$  is sufficient for the attained prediction error, then it seems that it does not explain the empirical success of the *deep* neural network as argued in the abstract.
> > >
> > > In the abstract, we write: *While deep neural networks (DNNs) have demonstrated remarkable empirical success in high-dimensional regression, their theoretical behavior in settings involving infinite-dimensional covariates remains largely unexplored.* We did not intend to claim empirical success of DNNs specifically for regression models with functional covariates. Rather, our motivation is that DNNs have shown strong performance in high-dimensional settings for learning low-dimensional structure in nonparametric regression. A natural question is therefore how neural-network-type estimators behave when the covariate is infinite-dimensional.
> > >
> > > In this paper, we use neural-network-type estimators as a new tool for studying regression with functional covariates. This is already a challenging problem because of vanishing eigengaps and the inverse nature of the problem. Existing methods often require either ad hoc estimators, such as the plug-in estimators in [1,2], or specialized techniques such as the change-of-measure argument in [3]. Thus, the theoretical contribution of this paper should not be interpreted as explaining the prediction power of DNNs in full generality, but rather as building a framework and technical tools for using neural-type estimators in regression models with functional covariates. We show that, in both linear and generalized linear settings, the DNN-type estimator achieves the same convergence rates as classical nonparametric ad hoc estimators.
> > >
> > > Moreover, many papers study the theoretical properties of shallow networks because their simpler structure helps capture and understand the behavior of such estimators, and they serve as a foundation for more complex cases. For example, [4] and [5] use two-layer neural networks to study the double-descent phenomenon in modern machine learning. In this sense, both these papers and our manuscript treat neural networks as a new class of estimators and develop tools to study the theoretical properties of a specific problem.
> > >
> > > > In Section 4, the authors consider a DNN with only 2 hidden layers, why not consider a deeper network?
> > >
> > > In the simulation studies, we did try different numbers of hidden layers, such as 3, 4, and 5, but the results were almost the same as, or only slightly different from those with 2 hidden layers. This is also consistent with the theoretical findings of the paper. We will provide more results on how depth affects the behavior in the Supplement when we revise the paper.
> > >
> > > Sincerely,
> > >
> > > The Authors
> > >
> > > [1] Cai, T. T., & Hall, P. (2006). *Prediction in functional linear regression*. The Annals of Statistics, 34(5), 2159-2179.
> > >
> > > [2] Hall, P., & Horowitz, J. L. (2007). *Methodology and convergence rates for functional linear regression*. The Annals of Statistics, 35(1), 70-91.
> > >
> > > [3] Dou, W. W., Pollard, D., & Zhou, H. H. (2012). *Estimation in functional regression for general exponential families*. The Annals of Statistics, 40(5), 2421-2451.
> > >
> > > [4] Mei, S., & Montanari, A. (2022). *The generalization error of random features regression: Precise asymptotics and the double descent curve*. Communications on Pure and Applied Mathematics, 75(4), 667-766.
> > >
> > > [5] Mei, S., Misiakiewicz, T., & Montanari, A. (2022). *Generalization error of random feature and kernel methods: Hypercontractivity and kernel matrix concentration*. Applied and Computational Harmonic Analysis, 59, 3-84.

---

### Official Review · Reviewer_ivfF · 2026-03-06

**Soundness:** 3
**Presentation:** 2
**Significance:** 3
**Originality:** 2
**Overall Recommendation:** 4
**Confidence:** 4

**Summary:**

This paper focuses on the theoretical properties of deep neural network (DNN) estimators for scalar-on-function regression. The paper studies the topic under three main settings: Functional linear regression, functional nonlinear regression (with known link function), and functional nonparametric regression (with unknown link function). The main contributions are the theoretical insights on establishing optimum convergence rates for the first two settings (i.e. functional linear and nonlinear regression), and empirical results demonstrating the trade-off between resolution and model efficiency. The results show that for linear and nonlinear regression with known link functions, similar convergence rates can be achieved with DNNs to those of classical estimators; and optimum performance is achieved (during dimensionality reduction) when fine local features are preserved while efficiency (i.e. avoiding curse of dimensionality) is maintained.

**Compliance With Llm Reviewing Policy:**

Affirmed.

**Final Justification:**

Main concerns were about Sec 3.3, and they are addressed in the rebuttal and the authors have committed to include the corresponding changes in the revised version.

**Key Questions For Authors:**

1. Could the authors clarify their contributions (differentiating themselves from existing work and highlighting the novelty) regarding fully nonparametric regression (corresponding mainly to Sec 3.3)?

**Limitations:**

yes

**Strengths And Weaknesses:**

Strengths:
* The submission is overall technically sound. Their approach for functional regression is methodologically appropriate, the claims on achieving optimal convergence for generalized functional models with known link functions are theoretically supported.
* The submission is overall well-written and easy to follow. The paper positions itself in the context of existing literature by clearly stating that they study theoretical performances of DNNs for functional regression, and extend existing work to infinite dimensional and non-compact domains. The methodology is structured around three main settings, which are logical to follow (however please see Weaknesses for Sec. 3.3. Functional Nonparametric Regression).
* The paper addresses an important and relevant problem since DNNs are widely used for regression tasks, however they are not theoretically well studied, especially regarding high dimensional functional covariates. The paper could influence or motivate future work on this topic, especially towards larger empirical studies that builds up on their numerical studies or fully nonparametric settings.
* Especially regarding the first two settings, the work provides some new insights on theoretical properties of DNNs for regression with functional covariates, and establishes theoretical grounds for the empirical deployment of such methods, supported by some empirical studies.

Weaknesses:
* While the empirical results overall support the claims of an optimum point for dimensionality reduction, Sec. 4 (Numerical Studies) has various weaknesses. It is not clearly and systematically communicated how the empirical results connect to and support different contributions of Sec. 3. The choices for experimental design are not clearly motivated (e.g. choice of datasets, choice of evaluation tasks as even though previous sections focus on regression, numerical studies include classification).
* Sec. 3.3. on nonparametric regression does not state clear technical contributions, but mostly focuses on existing literature and research directions. Some of the content of this section could be better fitting for other sections (e.g. Sec. 2 Background and literature review or Discussion and Conclusion). Moreover, Sec. 4 (Numerical Studies) could benefit from more details regarding reproducibility such as introducing metrics in more detail, implementation details of GLM, and further discussion on empirical results and outlook. Lastly, the authors could consider adding a Conclusion section. Minor comment: Proper referencing is needed for datasets instead of URLs (Column 2 L351-353 and L373-375).
* In Sec. 3.3 Functional Nonparametric Regression, the paper does not provide concrete ideas or contributions for future work to build up on.
* Contributions of Sec 3.3. are not clearly distinguished from related literature. The section does not provide new ideas or insights for this setting, or they are not clearly highlighted.

---

> ### Author Rebuttal · Authors · 2026-03-28
>
> Response to Reviewer ivfF
>
> We thank the reviewer for the careful reading and constructive comments. We are glad the reviewer finds the paper technically sound. In the revision, we will revise Sections 3.3 and 4 as suggested, clarify the purpose of each experiment, distinguish our contributions from related work, add a brief conclusion, and add datasets references.
>
> > Numerical studies are not systematically connected to Section 3
>
> Section 4 is meant to illustrate the theory in Section 3. In practice, functional covariates are observed on a finite grid and used as finite-dimensional inputs; with a dense grid. When the observation grid is sufficiently dense, the finite-dimensional approximation is expected to closely reflect the theoretical setting in which the full function is observed. Our synthetic experiments therefore mirror the assumptions behind the convergence results. We study how prediction error scales with both sample size and the number of observation points per curve, and plot $\log(MSE)$ versus $\log(n)$ so the slope gives an empirical check of the predicted rate. For real datasets, the true regression function is unknown, so these experiments are not intended to verify theorems directly, but to illustrate practical implications of the theory, including the effect of number of observations and the trade-off in choosing $K_n$. In the revision, we will restructure the beginning of Section 4 to highlight these connections more explicitly.
>
> > Experimental design choices are insufficiently motivated; proper referencing is needed for datasets; classification seems inconsistent with the earlier regression focus.
>
> We chose the UK electricity and NHANES datasets because they are standard benchmarks in the functional-data and longitudinal-data literature; we will add formal citations and explain this choice explicitly in Section 4. The classification experiments are also fully consistent with our theory: they are a special case of the generalized functional regression model in Section 3.2, where $Y_i\mid X_i$ belongs to the exponential family. Binary responses correspond to the Bernoulli case, i.e., functional logistic regression. We will revise the text to make this explicit.
>
> > Section 3.3 does not clearly state technical contributions; some material would fit better in Background or Discussion. Section 3.3 also does not clearly distinguish its contribution from related work or suggest future directions.
>
> We appreciate this suggestion and will rewrite the start of Section 3.3 to state its contribution directly. The main result is a lower bound for fully nonparametric functional regression over a H\"older class, showing that DNN estimators can achieve at best logarithmic pointwise rates; polynomial rates are therefore impossible without additional structural assumptions. We will move part of the conceptual discussion to Background/Discussion and make clearer that Scenario (3) plays a complementary role to Scenarios (1)-(2): the latter are benchmark settings where optimal polynomial rates are achievable, while Scenario (3) characterizes the boundary of what is statistically possible in the fully nonparametric case. Thus, the slower rate in Scenario (3) is not a weakness of DNNs or our proofs, but an intrinsic limitation of the problem.
>
> We will also clarify what is new in this subsection. First, the lower bound is tailored to the growing-dimension truncation representation used by DNN estimators for functional inputs, and explicitly captures the resulting logarithmic barrier. Second, it explains why fixed-dimensional DNN theory does not circumvent this limitation: the truncation dimension $K_n$ must grow with $n$ (typically polynomially), so fixed-dimensional arguments do not yield polynomial rates here. This identifies a concrete direction for future work: finding structural conditions under which the logarithmic barrier can be overcome.
>
> > Section 4 needs more reproducibility detail; consider adding a Conclusion.
>
> We will expand Section 4 accordingly. We report MSE for regression and AUC for classification on held-out test sets. For the GLM baseline, we use `sklearn.linear_model.LogisticRegression` with `penalty=None` and `max_iter=3000`; other parameters are left at default values. We will add fuller details on data splitting, preprocessing, and tuning so the experiments can be reproduced.
>
> We will also add a short conclusion emphasizing that DNN estimators attain minimax-optimal polynomial rates for functional linear and generalized linear models, while in fully nonparametric functional regression we prove a lower bound showing that polynomial rates are generally unattainable without additional structure. Overall, the paper highlights both the promise of DNNs in structured functional models and the fundamental barriers that arise in the general case.

---

> > ### Author Rebuttal · Reviewer_ivfF · 2026-04-02
> >
> > Thank you for the clarifications. My concerns have been addressed. I update my score.

---

> > > ### Author Response · Authors · 2026-04-02
> > >
> > > Dear Reviewer ivfF,
> > >
> > > Thank you very much for your thoughtful feedback and for adjusting the score. We sincerely appreciate your time and your recognition of our responses.
> > >
> > > Sincerely,
> > > The Authors

---

### Official Review · Reviewer_i5jY · 2026-03-11

**Soundness:** 3
**Presentation:** 3
**Significance:** 2
**Originality:** 2
**Overall Recommendation:** 5
**Confidence:** 4

**Summary:**

This paper investigates the theoretical properties of Deep Neural Network (DNN) estimators for scalar-on-function regression. While DNNs have seen immense empirical success and have been studied extensively for scalar-on-vector regression, similar exploration is limited for function-valued regression. The main purpose of this paper is to address this gap by providing rates of convergence for scalar-on-function regression when the regression function is fit using a DNN. For functional linear & generalized linear models, the authors demonstrate that DNN estimators achieve minimax-optimal polynomial convergence rates (up to logarithmic factors) for models where the slope functions admit a polynomially decaying expansions along the eigenfunctions of the functional covariates. These results match classical FDA benchmarks under less stringent assumptions compared to most existing results. Under a fully nonparametric functional regression setup and Holder smoothness and Gaussian process predictors, the paper establishes a minimax lower bound proving that polynomial rates are fundamentally unattainable in the general nonparametric setting and instead, only logarithmic rates are possible. In terms of technical contributions, the analysis bridges finite-dimensional approximation theory with the growing-dimensional regime required by functional principal component truncation. Furthermore, the proofs account for non-compact functional inputs, extending the reach of standard DNN theory. Finally, the authors complement the theoretical findings with empirical simulations.

**Compliance With Llm Reviewing Policy:**

Affirmed.

**Final Justification:**

The authors have adequately answered my questions. I have updated my score.

**Key Questions For Authors:**

*The paper only considers observing whole functional predictors rather than discretely sampled versions of them, which have many additional subtleties. Can the authors comment on what can be expected given existing literature on this?
*  For functional linear regression, the paper only considers the polynomial decay of slope coefficients -- can the authors comment on possible other cases of summable coefficients?
* How to adaptively choose K?
* What are matching upper bounds for the nonparametric case

**Limitations:**

yes

**Strengths And Weaknesses:**

Strength:
*The paper successfully bridges a gap in using DNN for scalar on function regression using DNN for functional linear and generalized linear models.
* Technically, the results of the paper need to extend some of the earlier theory and are minimax optimal upto log factors

Weakness
* The paper only considers observing whole functional predictors rather than discretely sampled versions of it which has many additional subtleties
* The nonparametric part of the paper only provides a lower bound and lacks a discussion on its sharpness
* The paper does not discuss adaptive choice of truncation K

---

> ### Author Rebuttal · Authors · 2026-03-28
>
> Dear Reviewer i5jY,
>
> Thank you for your constructive review and thoughtful comments. We appreciate your positive assessment of our work. Below we respond to each point.
>
> > The paper only considers observing whole functional predictors rather than discretely sampled versions of it which has many additional subtleties.
>
> Thank you for raising this important question. In functional data analysis, the effect of discrete observation schemes on convergence rates is well understood for mean and covariance estimation, but is much less understood for regression with functional covariates. The main difficulty comes from the infinite-dimensional nature of the problem. In regression, one must handle an inverse problem, which requires sharp bound of eigenvalues and eigenfunctions with diverging indices. Some recent works have made progress in this direction. For example, [1] studies least-squares estimation in functional linear regression, and [2] establishes optimal convergence rates for eigenvalues and eigenfunctions at diverging indices, making it possible to analyze plug-in estimators such as [3,4]. However, these analyses are generally require highly problem-specific and technically delicate arguments
>
> Because regression with discretely observed functional covariates is already challenging, and because the theory of DNN estimators is still not fully understood even in the fully observed setting, in this paper we focus on the fully observed case as a necessary first step. Our method depends only on estimated FPCA scores, so in principle it could be combined with existing score-estimation methods for discretely observed functional data, such as [1]. That said, obtaining optimal convergence rates would likely require sufficiently dense observations per subject, since the additional score-estimation error must be sharply controlled. We therefore leave this as an important direction for future work.
>
> > Only the polynomial decay of slope coefficients is considered.
>
> The assumption $b_k\asymp k^{-b}$ is only a convenient benchmark class widely used in the functional data literature. In the proof of Theorem 3.4, this assumption is used only to bound the truncation bias $\sum_{k>K}\lambda_k b_k^2$, where $\beta(t)=\sum_{k\ge1} b_k\phi_k(t)$. Therefore, the same argument extends to general square-summable coefficient sequences, with the prediction error governed by the weighted tail $\sum_{k>K}\lambda_k b_k^2$, rather than specifically by polynomial decay of $b_k$. More specifically, up to the same logarithmic factor as in Theorem 3.4, our proof yields
> $$\mathfrak R(\hat f,f)\lesssim \frac{K_n}{n}+\sum_{k>K_n}\lambda_k b_k^2.$$
> Thus, the polynomial case is simply a special case that gives an explicit rate, namely $\sum_{k>K}\lambda_k b_k^2\asymp K^{1-a-2b}$. This also clarifies other regimes. If one only assumes $\sum_{k\ge1} b_k^2\le B^2$, then $\sum_{k>K}\lambda_k b_k^2\le B^2\lambda_{K+1}\asymp K^{-a}$, leading to the generic rate $n^{-a/(a+1)}$ up to logarithmic factors. If the coefficients decay exponentially, one obtains a nearly parametric prediction rate up to logs; if $\beta$ is finite-rank, one obtains a parametric $n^{-1}$-type rate up to logs. We will add a remark after Theorem 3.4 to clarify that the polynomial assumption is imposed mainly to present an explicit benchmark rate, while the proof itself applies more generally.
>
>
> > How to adaptively choose $K$?
>
> Theoretically, there is an optimal order of $K$ that balances approximation bias and estimation variance. In practice, however, $K$ is usually selected in a data-driven way. In regression task, a common approach is predictive cross-validation, choosing $K$ to minimize out-of-sample prediction error. Another approach is the FVE criterion, for example taking the smallest K such that the leading K principal components explain a prescribed proportion of the total variation.
>
> > What are matching upper bounds for the nonparametric case?
>
> As discussed in Section 3.3, by using existing results from [5], one can obtain an upper bound of order $\log n$ for functional nonparametric regression. Since this conclusion follows relatively directly from the available theory, we did not state it as a separate theorem or proposition. Instead, our intention there is to emphasize the limitation side of the problem, namely that without additional structural assumptions one cannot in general expect polynomial rates in fully nonparametric functional regression.
>
>
>
> [1] Zhou, H., Yao, F., & Zhang, H. (2023). Biometrika, 110(2), 381-393.
>
> [2] Zhou, H., Wei, D., & Yao, F. (2025).  The Annals of Statistics, 53(5), 2103-2127.
>
> [3] Cai, T. T., & Hall, P. (2006). The Annals of Statistics, 34(5), 2159-2179.
>
> [4] Hall, P., & Horowitz, J. L. (2007).  The Annals of Statistics, 70-91.
>
> [5] Schmidt-Hieber, J. (2020).  The Annals of Statistics, 48(4), 1875-1897.

---

> > ### Author Rebuttal · Reviewer_i5jY · 2026-04-03
> >
> > The authors have adequately answered all my comments, and I have no further questions.

---

> > > ### Author Response · Authors · 2026-04-04
> > >
> > > Dear Reviewer i5jY,
> > >
> > > We thank you again for your thoughtful review and reply. We sincerely appreciate your time and your recognition of our work.
> > >
> > > Sincerely,
> > >
> > > The Authors

---

### Official Review · Reviewer_u6MX · 2026-03-14

**Soundness:** 4
**Presentation:** 4
**Significance:** 4
**Originality:** 3
**Overall Recommendation:** 5
**Confidence:** 5

**Summary:**

This paper considers a scare on function regression model where the regression function between the functional covariates and the scalar response is modeling  using a deep neural network (specially, a feedforward architecture  on the principle components scores). Three models are considered: functional linear regression, generalized functional linear regression and a nonparametric functional regression model. Under the three models, the authors show that the corresponding estimators (by minimizing an appropriate  objective function) achieve minimax rates.

**Compliance With Llm Reviewing Policy:**

Affirmed.

**Key Questions For Authors:**

The paper essentially shows that DNN based estimators can perform as well as the classical estimators  by achieving the minimax rates (e..g kernel based estimators). Are their scenarios that DNN based estimators can outperform the classical estimators?

**Limitations:**

None. An overall solid contribution to ICML.

**Strengths And Weaknesses:**

Strengths:

+A well-written paper with a clear and through discussion on the theory of functional regression literature.

+strong theoretical results for the DNN based estimators  with a careful discussion on the results and corresponding proof techniques

---

> ### Author Rebuttal · Authors · 2026-03-28
>
> Dear Reviewer u6MX,
>
> Thanks for your careful review and encouraging comments. We sincerely appreciate your positive assessment of our work and your insightful question. We think an important advantage of the DNN framework in regression with functional covariates is its generality. It provides a unified approach for learning the regression function across a broad range of model structures, rather than requiring a separate estimator tailored to each specific setting.
>
> For classical models such as functional linear regression, functional generalized linear models, and functional Cox models, specialized estimators have indeed been developed by exploiting the underlying model structure.
> For example, in functional linear regression, one can construct an estimator based on the normal equations, which substantially simplifies the technical analysis [1,2].
> In functional generalized linear models, the normal equation no longer exists and classical approaches typically require a change-of-measure argument in order to derive the optimal convergence rate [3].
> The advantage of the DNN framework is that it provides a single flexible procedure for estimating regression functions with functional covariates in both structured and unstructured settings, together with a unified framework for theoretical analysis.
>
> This is exactly the role of scenarios (1)--(3). Scenarios (1) and (2) show that, in the linear and generalized linear settings, the DNN estimator does not lose statistical efficiency: it still achieves the optimal rates known from the classical literature. Thus, adopting a generic DNN procedure does not incur a penalty in these structured benchmark models. Scenario (3), on the other hand, shows that without further assumptions, polynomial rates are not attainable in fully nonparametric functional regression, which is consistent with classical nonparametric theory. Hence, the slower rate there reflects the intrinsic difficulty of the problem rather than any weakness of the DNN approach.
>
> At the same time, the DNN framework becomes particularly valuable when additional structure is imposed beyond the classical linear setting. For instance, if the regression function $f$ has a compositional structure built from low-dimensional H{\"o}lder continuous components, or if $f$ belongs to an anisotropic Besov space, then polynomial rates can again be achieved. These richer nonlinear structures are difficult to handle within traditional model-specific procedures, whereas they fit naturally into the approximation framework of deep networks. In this sense, the DNN approach not only unifies the structured cases already covered by classical estimators, but also extends naturally to important nonlinear functional regression settings that classical estimators are not designed to address.
>
>
> [1] Cai, T. T., & Hall, P. (2006). Prediction in functional linear regression. The Annals of Statistics, 34(5), 2159-2179.
>
> [2] Hall, P., & Horowitz, J. L. (2007). Methodology and Convergence Rates for Functional Linear Regression. The Annals of Statistics, 70-91.
>
> [3] Dou, W. W., Pollard, D., and Zhou, H. H. Estimation in functional regression for general exponential families. Annals of Statistics, 40(5) 2421-2451, 2012.

---

> > ### Author Rebuttal · Reviewer_u6MX · 2026-04-05
> >
> > I thank the authors for the response.

---

> > > ### Author Response · Authors · 2026-04-05
> > >
> > > Dear Reviewer u6MX,
> > >
> > > We thank you again for your thoughtful review and reply. We sincerely appreciate your time and your recognition of our work.
> > >
> > > Sincerely,
> > >
> > > The Authors

---

### Decision · Program_Chairs · 2026-04-30

**Decision:**

Accept (regular)

**Comment:**

This paper studies deep neural network regression with functional covariates and develops theoretical guarantees in several structured and unstructured regimes. The manuscript is technically meaningful and stronger than average on theory, and most reviewers were positive after discussion.

The main reservation is that the paper's broad DNN motivation is not fully matched by its formal claims. In particular, one reviewer remained unconvinced that the manuscript really explains the benefit of DNN, as opposed to showing that neural estimators can recover classical rates in certain settings. I think that is a fair limitation, but not one that must force rejection. I therefore recommend Weak accept.